# Embedding Hybrid Systems into Continuous Latent Vector Fields

**Sangli Teng** [1]   **Hang Liu** [2]   **Koushil Sreenath** [1]

## Abstract

This work proves that an $n$-dimensional hybrid system can be embedded into an $m$-dimensional Euclidean space equipped with a continuous vector field on its embedded image whenever $m > 2n$. This result suggests that an *intrinsically* discontinuous hybrid system generically admits a continuous *extrinsic* representation that is well-posed for differentiable optimization. Building on this existence theorem, we show that a latent Neural ODE with consistency loss in both the latent and state space can accurately recover the flow of hybrid systems. Extensive experiments suggest the proposed method outperforms the existing method in learning hybrid systems with varying geometries from only time series data.

## 1. Introduction

Hybrid systems (hybrid automata) model a broad class of physical (Westervelt et al., 2003; Posa et al., 2014) and cyber-physical processes (Tabuada, 2007; Ames et al., 2014) by combining continuous-time vector fields with discrete state resets. Despite its powerful expressiveness, the hybrid system has nonsmooth or discontinuous state evolution that is not well-posed for differentiable optimization (Paszke et al., 2017), especially at state resets.

To learn hybrid systems from data, traditional methods (Poli et al., 2021; Liu et al., 2025) partition the trajectories into different segments and learn the dynamics in each mode. However, these methods require mode selection that is combinatorially complicated. On the other hand, the event function-based method (Chen et al., 2020) tries to differentiate through the state resets, which, however, suffers from ill-conditioning and bad initializations.

A more recent structural alternative is to represent the hybrid dynamics as continuous flows. The hybrid system the-

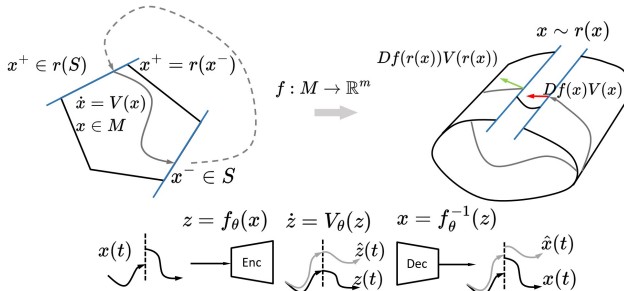

Figure 1: We proved that the $n-$dimensional *discontinuous* flow of a hybrid system can be embedded into a latent space equipped with an $m-$dimensional *continuous* extrinsic vector field when $m > 2n$. The latent embedding can be learned by the proposed latent ODE framework CHyLL++.

ory (Simic et al., 2005) suggests that the state reset functions induce an equivalence relationship to *glue* the partitioned state space into a continuous latent manifold. Furthermore, the glued manifold can be reconstructed from time-series data in our previous **C**ontinuous **Hy**brid System **L**earning in **L**atent Space  (CHyLL) algorithm  (Teng et al., 2025), leveraging the Whitney Embedding Theorem (Hirsch, 2012).

However, the results in (Simic et al., 2005; Teng et al., 2025) do not suggest that the *latent vector field* is continuous. To further improve the differentiability, we pose the question: ***"Do hybrid systems admit provably continuous latent embedding that induces a continuous vector field?"***. Answering this question will lay a foundation to make differentiable optimization well-posed for hybrid system learning. As in Figure 1, we make the following contributions:

**Contribution:** The ***theoretical*** contribution is to prove that an $n$-dimensional hybrid system can be embedded into an $m$-dimensional Euclidean space equipped with a continuous vector field on its embedded image as long as $m > 2n$. This theorem suggests that an *intrinsically* discontinuous hybrid system admits an *extrinsically* continuous representation that is well-posed for differentiable optimization. Based on this theorem, the ***algorithmic*** contribution is a latent Neural ODE framework for learning hybrid systems from time series data. The experiments and ablations show that the consistency loss in both the latent and the observation space is the key to accurately recovering the flow of hybrid systems with varying topologies. The implementation is included in https://github.com/SangliTeng/Continuous-Hybrid-System-Learning.

---

[1]University of California, Berkeley, CA, USA [2]University of Michigan, Ann Arbor, MI, USA. Correspondence to: Sangli Teng <sangliteng@berkeley.edu>.

*Proceedings of the $43^{rd}$ International Conference on Machine Learning*, Seoul, South Korea. PMLR 306, 2026. Copyright 2026 by the author(s).

## 2. Related Work

Hybrid systems can model a wide range of autonomous systems, such as legged locomotion (Westervelt et al., 2003; 2018), task and motion planning (Garrett et al., 2021), and serve as an ideal abstraction for verification of safety (Ames et al., 2019) or formal guarantees specified by temporal logic (Leung et al., 2023; Pan et al., 2023). Compared to conventional dynamical systems that can be represented by a single ordinary differential equation, hybrid systems are governed by both continuous-time ODE and discrete-time state transitions, which makes it expressive but more challenging for differentiable optimizations.

The Neural ODE (Chen et al., 2018) is a standard tool to learn the underlying vector field of the time-series observations. Though the universal approximation theorem suggests that any *continuous* function on a compact domain can be uniformly approximated by neural networks (Lipschitz continuous) (Kidger & Lyons, 2020; Cybenko, 1989; Hornik, 1991), the uniform approximation is not possible for discontinuous functions. However, the instantaneous state transitions in hybrid systems can only be represented by an impulse or discontinuous function, which is not well-posed for differentiable optimization. To mitigate this issue, the Neural Event ODE (Chen et al., 2020) differentiates the reset and event functions in an event-based simulation. However, a randomly initialized neural event function is generally ill-conditioned, making the simulation difficult to proceed. More recent work, such as (Liu et al., 2025) and (Poli et al., 2021) admits a mode selector to assign each mode a distinct continuous dynamics. However, this line of methods suffers from the combinatorial complexity in mode selections, and the number of modes may be unknown.

Other than learning the hybrid systems in the state space, the topology of hybrid systems indicates that they can be formulated as a continuous manifold (Simic et al., 2005). Building on this theorem, (Teng et al., 2025) represents the hybrid systems as a latent flow leveraging the Whitney Embedding Theorem (Hirsch, 2012) to obtain a singularity-free representation by increasing the latent dimension and a few geometry-based inductive biases.

In this work, we not only inherit continuity of the latent space of hybrid systems through its topology (Simic et al., 2005) and the Whitney embedding theorem (Hirsch, 2012; Teng et al., 2025), but further prove that the latent vector field can be continuous. Our main result builds on the transversality conditions (Abraham & Robbin, 1967), which are powerful tools for proving the generic properties of dynamical systems.

## 3. Preliminary

In this section, we discuss the preliminaries of differential geometry and a geometric description of hybrid systems.

### 3.1. Differential Geometry

Consider a finite-dimensional smooth manifold $M$. The tangent space at a point $x \in M$ is denoted by $T_x M$. The tangent bundle $TM := \bigcup_{x \in M} T_x M$ is the disjoint union of tangent spaces. A smooth vector field is a map $V : M \to TM$ such that $V(x) \in T_x M$ for all $x \in M$. The set of all smooth vector fields on $M$ is denoted by $\mathfrak{X}(M)$.

For any smooth curve $c : (-\varepsilon, \varepsilon) \to M$ satisfying $c(0) = x$ and $\dot{c}(0) \in T_x M$, the tangent map $Df(x) : T_x M \to T_{f(x)} N$ of the smooth map $f : M \to N$ is defined by: $Df(x)\dot{c}(0) = \frac{d}{dt}\big(f \circ c\big)(t)\big|_{t=0}$. For $f : M \to N$ and $g : G \to M$ with $G$ a smooth manifold, the tangent map of $f \circ g : G \to N$ satisfies the chain rule $D(f \circ g)(x) = Df\big(g(x)\big) \circ Dg(x)$.

We introduce two nondegenerate property for $f : M \to N$.

**Definition 1** (Immersion (Hirsch, 2012)). *$f$ is an immersion if the tangent map $Df : T_x M \to T_{f(x)} N$ is an injective function, or equivalently, $\operatorname{rank} T_x f = \dim M$.*

**Definition 2** (Embedding (Hirsch, 2012)). *$f$ is an embedding if $f$ is an immersion and $f$ maps $M$ homeomorphically[1] onto its image.*

For a manifold with boundaries, we can conduct our analysis in the collar coordinates:

**Theorem 1** ((Hirsch, 2012)). *For a manifold $M$ with boundary $S \subset M$, a submanifold of $M$ with co-dimension 1, there exists a collar embedding such that $x$ specifies the position on the boundary and $t$ denotes the inward components of the collar that points to the* interior *of $M$:*

$$\kappa_S(x, t) : S \times [0, \varepsilon) \to M \tag{1}$$

*with $\kappa_S(x, 0) = x, \forall x \in S$.*

### 3.2. Hybrid Systems

We consider a geometric description of the hybrid system (Clark & Bloch, 2023; Simic et al., 2005) as shown in Figure 1.

**Definition 3** (Hybrid System). *A hybrid system is a 4-tuple $\mathcal{H} = (M, S, V, r)$ with the following components:*

*(State Space) $M$ is an $n$-dim smooth manifold.*

*(Guard) $S \subset \partial M$ is a $(n-1)$-dim smooth submanifold.*

*(Dynamics) $V : M \to TM$ is a smooth vector field.*

---

[1] A function $f$ is a homeomorphism if it is bijective, continuous, and its inverse $f^{-1}$ is also continuous.

*(Reset)* $r\colon S \rightarrow r(S) \subset \partial M$ *a diffeomorphism onto its image and* $r(S)$ *is an embedded submanifold.*

The dynamics of $\mathcal{H}$ can be described as an event ODE:

$$\begin{cases} \dot{x} = V(x), & x \notin S, \\ x^+ = r(x^-), & x^- \in S. \end{cases} \quad (\mathcal{H}\text{-Dynamics})$$

We assume $\mathcal{H}$ has the following properties:

**Assumption 1.** *$M$ and $S$ are compact.*

**Assumption 2.** *$V(x)$ points* outward *along the interior of $x \in S$ and* inward *along the interior of $x \in r(S)$.*

**Assumption 3.** *$S \cap r(S) = \emptyset$ and the intersection of their closure $\overline{S} \cap \overline{r(S)}$ has co-dimension at least 2.*

Assumption 2 ensures the trajectories will traverse the boundary, and Assumption 3 avoids repeated state resets from happening instantaneously.

Though $S$ and $r(S)$ are different and possibly disconnected, $r(\cdot)$ is a diffeomorphism that defines an equivalence relationship that "glues" $S$ and $r(S)$ to reformulate the state space $M$ as a continuous manifold as indicated in Figure 1:

**Theorem 2** (Hybrifold (Simic et al., 2005)). *Let $\sim$ be the equivalence relation by $x \sim r(x), \forall x \in S$. We collapse the equivalence class by $\sim$ to a point to obtain the piecewise smooth topological manifold, namely, the hybrifold:*

$$M_{\mathcal{H}} = M/\sim, \quad (\text{Hybrifold})$$

### 3.3. Generic Property and Transversal Maps

Denote the $k$-time continuous functions from a finite-dimensional manifold $M$ to $N$ as $C^k(M, N)$. The main goal of this work is to explore if a certain property is held for the majority of elements in $C^k(M, N)$. Thus, we explore the "generic property", and informally, we have:

**Remark 1** (Generic property). *A property is generic if it holds in a "large set" [2] that contains "most" of the elements in a topological space. For example, the set of irrational numbers $\mathbb{R} \setminus \mathbb{Q}$ contains "most" of the elements in $\mathbb{R}$.*

For more regularity conditions of $C^k(M, \mathbb{R}^m)$ space, please refer to Section A.2, A.3, and A.4.

To study the generic property of $f \in C^k(M, N)$, we leverage the tools related to the transversal map:

**Definition 4** (Transversality Condition (Hirsch, 2012)). *Let $M, N$ be smooth finite-dimensional manifolds and $Z \subset N$ an embedded submanifold. A $C^k (k \geq 1)$ map $f : M \rightarrow N$ is* transverse *to $Z$, written $f \pitchfork Z$, if for every $x \in f^{-1}(Z)$,*

$$\text{Im}(df(x)) + T_{f(x)}Z = T_{f(x)}N. \quad (\text{Transversality})$$

---
[2] A residual set, i.e., the complement of a countable union of nowhere dense sets.

**Theorem 3** (Preimage Theorem under Transversality (Abraham & Robbin, 1967)). *If $f \pitchfork Z$, then $f^{-1}(Z)$ is an embedded submanifold of $M$ and with dimension:*

$$\dim f^{-1}(Z) = \dim M - \text{codim}_N Z. \quad (2)$$

In our proof shown in the following sections, we construct the set we wish to avoid as the manifold $Z$, and we let $\dim f^{-1}(Z) < 0$ to ensure the image of $f$ is empty (bad things never happen).

**Theorem 4** (Parametric Transversality Theorem, Section 19.1 (Abraham & Robbin, 1967) ). *Let $M, N$ be $C^k$ finite-dimensional manifolds, and let $Z \subset N$ be a closed $C^k$ submanifold. Let $\mathcal{P}$ be a $C^k$ Banach manifold (e.g., $C^k(M, N) : M \rightarrow N$). Define the total map $\Gamma : M \times \mathcal{P} \rightarrow N$ and the associated slice $\Gamma(\cdot \mid p) : M \rightarrow N, \Gamma(x \mid p) := \Gamma(x, p)$. Assume the following conditions hold:*

*(i) The total map $\Gamma$ is of class $C^k$;*

*(ii) $\Gamma \pitchfork Z$ (the total map is transversal to $Z$);*

*(iii) $r > \max(0, \dim M - \text{codim}_N Z)$.*

*Then the set of parameters $\mathcal{P}_Z = \{ p \in \mathcal{P} : \Gamma(\cdot \mid p) \pitchfork Z \}$ is residual (and hence dense) in $\mathcal{P}$.*

As a consequence of Theorem 4, one can show that the following embedding exists generically:

**Theorem 5** (Whitney Embedding Theorem (weak) (Hirsch, 2012)). *Any $C^k$-manifold $M$ ($k \geq 1$) of dimension $n$ can be embedded into $\mathbb{R}^m$ if $m > 2n$.*

## 4. Problem Formulation

We consider the problem of learning the hybrid system from time series data:

**Problem 1.** *Consider the flow of $\mathcal{H}$ recorded as dataset $\mathcal{X} := \{(t_0, x_0), (t_1, x_1), (t_2, x_2), \cdots, (t_T, x_T)\}$. Our goal is to recover the flow of $\mathcal{H}$ from $\mathcal{X}$.*

We note that due to the reset map $r$ on the guard surface $S$, the flow of $\mathcal{H}$ is discontinuous and extremely hard to differentiate. Although Theorem 5 suggests that (Hybrifold) employs a continuous representation to make the flow of $\mathcal{H}$ continuous in a learned latent space (Teng et al., 2025), the vector field on (Hybrifold) is not guaranteed to be continuous. Thus, we ask:

**Problem 2.** *Can we embed $\mathcal{H}$ in a latent space and characterize the flow of $\mathcal{H}$ by a continuous latent vector field?*

By doing so, we can convert $\mathcal{H}$ into a globally continuous structure, which will significantly improve its differentiability for machine learning. In this work, we prove that such a latent representation can be constructed by elements that **generically exist** in the space of functions.

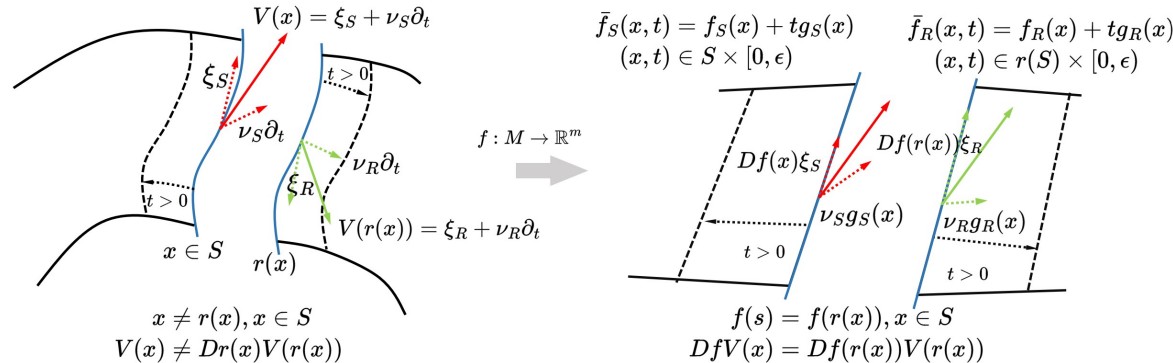

Figure 2: **Left:** the state and vector field at pre-impact state $x \in S$ and the post-impact state $r(x) \in r(S)$ are possibly mismatched as an *intrinsic* property of $\mathcal{H}$. **Right:** by designing the embedding $f : M \to \mathbb{R}^m$, the *extrinsic* representation satisfy (C-1) and (C-2). The additional degree of freedom by $f$ is the key for the extrinsic representation to eliminate the discontinuity.

## 5. Main Result

In this section, we leverage Theorem 3 and 4 to show that whenever $m > 2n$, there generically exists a continuous function (encoder) $f : M \to \mathbb{R}^m$ to make the latent manifold and its induced vector field both continuous on the entire $M$. Formally, we have:

**Theorem 6** (Continuous Extrinsic Representation of Hybrid Systems). *For a hybrid system $\mathcal{H}$ satisfying Assumption 1, 2 and 3, whenever $m > 2n$, there exists an $f \in C^k(M, \mathbb{R}^m)$ that satisfies:*

$$f(x) = f(r(x)), \forall x \in S \qquad \text{(C-1)}$$

$$Df(x)V(x) = Df(r(x))V(r(x)), \forall x \in S \qquad \text{(C-2)}$$

$$f \text{ induces an embedding of } M_{\mathcal{H}} \text{ into } \mathbb{R}^m. \qquad \text{(C-3)}$$

*Moreover, $f$ can be obtained by a generic choice within the admissible construction space.*

*Proof.* The full proof is in Section B, and we here briefly introduce the procedure. The proof is to show that a construction of $f$ exists *generically* when $m > 2n$ leveraging Theorem 4. The construction of $f$ is done on the collar coordinates ensured by Theorem 1.

Denote $R := r(S)$. To enforce (C-1), we choose an embedding $g : S \to \mathbb{R}^m$ (guaranteed by Theorem 5) to embed $S$−side as $f_S = g$ and $R$−side as $f_R = g \circ r^{-1}$ which ensures $f_R(r(x)) = f_S(x), \forall x \in S$. Then we apply the first order Taylor extension in the collar coordinate on $S$−side as $\bar{f}_S(x,t) = f_S(x) + tg_S(x)$ with $\bar{f}_S(x,t) : S \times [\epsilon, 0) \to M$ defined on the collar coordinates with $f_S(x)$ the components on the boundary $S$ and $g_S : S \to \mathbb{R}^m$ the component inward $S$ (similarly on $R$-side). By choosing $g_S$ and enforcing (C-2), we can uniquely determine $g_R$ on $R$ side. Given this construction in the collar neighborhood $\text{Img}(\kappa_S) \cup \text{Img}(\kappa_R)$, we show that whenever $m > 2n$, a generic choice of $g$ and $g_S$ ensures the first-order extension

is injective and without rank deficiency. Finally, we extend the local property to the entire $M$.

The geometric interpretation of the theorem is illustrated in Figure 2 in the collar coordinates in $M$ and $f(M)$. ☐

Based on Theorem 6, we can confirm that the answer to Problem 2 is positive.

**Corollary 1** (Differentiability of latent flow). *Let $z(t) = f(x(t))$ be the latent trajectory given a generic choice of $f$ and $x(t)$ the hybrid execution of $\mathcal{H}$. By Theorem 6, the induced latent vector field $\dot{z} = Df(x)V(x)$ is $C^0$ on $f(M)$ and the the latent trajectory $z(t)$ is $C^1$ in time.*

*Proof.* This is an immediate result by (C-1) and (C-2) for the differentiability at any $x \in S \cup r(S)$, and (C-3) for the differentiability elsewhere. ☐

Though we have this *extrinsic* continuous representation $f$, we note that the *intrinsic* discontinuity of the hybrid system $\mathcal{H}$ is irrelevant to the representation.

**Remark 2** (Intrinsic and Extrinsic Representation of $\mathcal{H}$). *By (Simic et al., 2005), the* intrinsic *vector field of $\mathcal{H}$ is not guaranteed to be continuous as it is possible that $\exists x \in S$*

$$Dr(x)V(x) \neq V(r(x)).$$

*On the contrary, $f : M \to \mathbb{R}^m$ is the* extrinsic *representation that ensures (C-1) and (C-2) are satisfied on $S \cup r(S)$ by additional degree of freedom.*

In this work, we seek such an extrinsic representation that makes Problem 1 well-posed for differentiable optimization, despite the intrinsic non-smooth structures. Thus, we have the following learnable components of our latent ODE:

**Remark 3.** *Whenever $m > 2n$, a hybrid system $\mathcal{H}$ with hybrid dynamics ($\mathcal{H}$-Dynamics) can be embedded into a latent space by an encoder*

$$f_\theta : M \to \mathbb{R}^m, \qquad \text{(Encoder)}$$

*that satisfies $f_\theta(x) = f_\theta(r(x)), \forall x \in S$, and the associated latent vector field*

$$V_\theta : \text{Img}(f_\theta) \to \mathbb{R}^m, \qquad \text{(Latent Vector Field)}$$

*that relates to $V(\cdot)$ by $V_\theta(f_\theta(x)) = Df_\theta V(x), \forall x \in M$. To recover $x$ from $z \in \text{Img}(f_\theta)$, we have the the decoder:*

$$f_\theta^{-1} : \text{Img}(f_\theta) \to M. \qquad \text{(Decoder)}$$

*For machine learning implementations, we can optimize the parameter $\theta$ to obtain $f_\theta$, $V_\theta$, and $f_\theta^{-1}$ given $\mathcal{X}$, the time-series observation from the flow of $\mathcal{H}$.*

## 6. Learning $\mathcal{H}$ by Latent Neural ODE

Based on Remark 3, we propose a latent ODE framework that recovers the flow of $\mathcal{H}$ from time series data observation $\mathcal{D}$ by designing the encoder $f_\theta : M \to \mathbb{R}^m$, latent vector field $V_\theta : \text{Img}(f_\theta) \to \mathbb{R}^m$, and the decoder $f_\theta^{-1} : \text{Img}(f_\theta) \to \mathbb{R}^m$. Each of these components can be represented by an MLP.

The latent flow is obtained by integrating $V_\theta$ from the initial state $z_0 = f_\theta(x_0)$ by Neural ODE (Chen et al., 2018):

$$\hat{z}_k = \int_{t_0}^{t_k} V_\theta(z(t))dt + f_\theta(x_0). \qquad (3)$$

Then we enforce the consistency loss both in the state space:

$$\mathcal{L}_x = \text{MSE}(f_\theta^{-1}(\hat{z}_k), x_k), \qquad (4)$$

and in the latent space

$$\mathcal{L}_z = \text{MSE}(\hat{z}_k, f_\theta(x_k)). \qquad (5)$$

Though representing the latent vector field $V_\theta(x)$ as an MLP (with a finite Lipschitz constant (Virmaux & Scaman, 2018)) automatically ensures the learned latent flow is unique (Khalil, 2008) and continuous, we consider additional inductive bias as in (Teng et al., 2025) to explicitly enforce the conditions in Theorem 6 and see if they can further improve the performance.

To enforce (C-1), we have the gluing loss (Teng et al., 2025):

$$\mathcal{L}_g = \text{MSE}(f_\theta(x_k), f_\theta(x_{k+1})), \forall k \in \mathcal{I}, \qquad (6)$$

with $\mathcal{I}$ the index set that labels all the pre- and post-reset states by thresholding the empirical Lipchitz constant of the data point in $\mathcal{D}$, i.e., $\|\frac{x_{k+1}-x_k}{t_{k+1}-t_k}\|$.

Then we explore if we need to enforce (C-2) that is not used in (Teng et al., 2025). Thus, we design the velocity compatibility loss:

$$\mathcal{L}_v = \text{MSE}(\dot{z}_k^-, \dot{z}_k^+), \forall k \in \mathcal{I}, \qquad (7)$$

---

**Algorithm 1** CHyLL++

---

**Require:** Trajectory dataset $\mathcal{D}$; gluing index set $\mathcal{I}$; curriculum $\{T_1 < \cdots < T_\ell\}$; steps per curriculum $S$.
1: **for** $\ell = 1, \ldots, L$ **do**
2:     **for** step $= 1, \ldots, S$ **do**
3:         $x_{0:T_\ell} \sim \mathcal{D}$       ▷ Sample mini-trajectories
4:         $\hat{z}_{0:T_\ell} \xleftarrow{(3)} x_0$       ▷ Encode and rollout
5:         $\hat{x}_{0:T_\ell} \xleftarrow{f_\theta^{-1}(\cdot)} \hat{z}_{0:T_\ell}$       ▷ Decode
6:         $\mathcal{L}(\theta) \xleftarrow{(4-9)} \{\hat{x}_k, \hat{z}_k\}_{k=0}^{T_l}$   ▷ Compute loss
7:         $\theta \leftarrow \theta - \eta \nabla_\theta \mathcal{L}(\theta)$     ▷ Update parameters
8:     **end for**
9: **end for**
      **return** $f_\theta, V_\theta$ and $f_\theta^{-1}$.

---

with the pre-reset velocity approximated by $\hat{z}_k^- = \frac{f_\theta(x_k)-f_\theta(x_{k-1})}{t_k-t_{k-1}}$ the backward finite difference and the post-reset one by $\hat{z}_k^+ = \frac{f_\theta(x_{k+1})-f_\theta(x_k)}{t_{k+1}-t_k}$ the forward version.

Finally, to avoid the latent space from collapsing, we enforce the latent space to have positive covariance by

$$\mathcal{L}_c = \sum_{i=1}^{m} \text{ReLU}(\Lambda - \text{Cov}(f_\theta(x_k)_i)), \qquad (8)$$

with $\Lambda$ the threshold for the latent covariance (Teng et al., 2025). Finally, we have the loss function as:

$$\mathcal{L}(\theta) = w_x \mathcal{L}_x + w_z \mathcal{L}_z + w_g \mathcal{L}_g + w_v \mathcal{L}_v + w_c \mathcal{L}_c, \quad (9)$$

with the weights $w_{(\cdot)}$. Finally, we consider a rollout curriculum to learn the trajectories from short to long segments, and we conclude our method in Algorithm 1.

## 7. Numerical Experiments

### 7.1. Analytical Example

We provide an analytical example to illustrate how an inherently discontinuous hybrid systems admits a continuous representation in the embedded space.

Consider a vector field on a 1-dimensional disconnected manifold with different vector fields on each partition:

$$V(x) = \begin{cases} 1, x \in [0,1) \\ 2, x \in [2,3) \end{cases} \quad r(x) = \begin{cases} x+1, x=1 \\ x-3, x=3 \end{cases} \quad (10)$$

We can verify that the two vector fields are not intrinsically continuous by the fact that the pre- and post-reset velocity $(Dr(x^-)V(x^-), V(r(x^-)))$ at $x^- = 1, 2$ are $(1, 2)$ and $(2, 1)$ respectively, which is not identical. As the $r(x)$ glues the state space as a circle, we construct $f$ by sinusoidal functions. Thus we consider the equivalence relationship

Table 1: We conduct the experiments five times for each case to compute the MSE (mean ± std.dev). For methods that exhibit numerical instability, divergence, or obvious failure in capturing the pattern, we report the qualitative results. The notation * indicates the results reported from (Teng et al., 2025). Though (Teng et al., 2025) works for the first four cases, it overfits to one modality of the input signal for the last case. The plots for all the cases can be seen in Section C.2.

| | Proposed (ReLu, $\mathcal{L}_{x,z,c}$) | (Teng et al., 2025) | (Chen et al., 2018) | (Rubanova et al., 2019) | (Lusch et al., 2018) | (Chen et al., 2020) |
|---|---|---|---|---|---|---|
| Bouncing Ball | **0.158** ± 0.0376 | 0.237 ± 0.0629* | Large Penetration | Diverge | Diverge | Ill-conditioned |
| Torus | **0.00367** ± 0.00166 | 0.0164 ± 0.00989* | Pattern Incorrect | Pattern Incorrect | Diverge | Ill-conditioned |
| Klein Bottle | **0.00587** ± 0.00461 | 0.0220 ± 0.00584* | Pattern Incorrect | Pattern Incorrect | Diverge | Ill-conditioned |
| Three-Link Walker | **0.0952** ± 0.00424 | 0.234 ± 0.0140* | 0.275 ± 0.00610* | 0.253 ± 0.0770* | Diverge | Ill-conditioned |
| 3D Bouncing Ball | **0.162** ± 0.0150 | Collapse to $z$-direction | 0.5244 ± 0.0707 | Diverge | Diverge | Ill-conditioned |

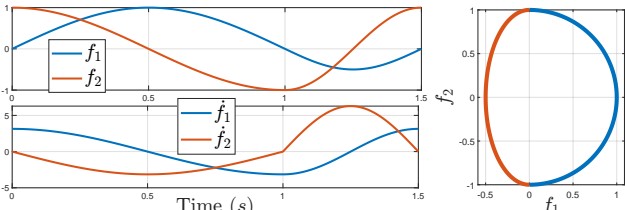

Figure 3: The 2-D embedding of a 1-D hybrid system. We find that the embedded manifold is continuous and admits a global $C^0$ extrinsic vector field $\dot{f}$. On the right-hand side, we see that the manifold is glued by hemispheres from two different ellipses.

$1 \sim 2$ and $3 \sim 0$ and analytically construct $f$ as:

$$f(x) = \begin{cases} A_1 y_1(x), x \in (0,1) \\ A_2 y_2(x), x \in (2,3) \end{cases}, \qquad (11)$$

with $y_1(x) = [\cos(\pi x), \sin(\pi x)]^\top$, $y_2(x) = [\cos(\pi(x-1)), \sin(\pi(x-1))]$ and $A_i \in \mathbb{R}^{2 \times 2}$ the linear coefficient matrices. Then we enforce (C-1) by letting $f(1) = f(2)$, $f(3) = f(0)$, and (C-2) by $Df(x^-) \circ V(x^-) = Df(r(x^-)) \circ V(r(x^-))$ with $x^- = 1, 3$. Thus, we have

$$A_1[-1,0]^\top = A_2[-1,0]^\top, \ A_1[1,0]^\top = A_2[1,0]^\top, \ (12)$$

for (C-1) and the following equations for (C-2):

$$A_1[0,1]^\top \pi = A_2[0,1]^\top 2\pi, \ A_1[0,1]^\top 2\pi = A_2[0,1]^\top \pi. \tag{13}$$

Thus, we choose a solution that also makes $f$ an embedding:

$$A_1 = \begin{bmatrix} 0 & 1 \\ 1 & 0 \end{bmatrix}, A_2 = \begin{bmatrix} 0 & -0.5 \\ 1 & 0 \end{bmatrix}.$$

Finally, we have the time evolutions of $f$ and $\dot{f}$ shown in Figure 3, which is a 1D continuous manifold embedded in 2D space Euclidean space equipped with a continuous vector field on it. Then, for each segment of the manifold, we can design a decoder by $\text{atan2}(\cdot, \cdot)$ with proper scaling of $f_1$ and $f_2$ as the inverse of the encoder $f$.

### 7.2. Learning $\mathcal{H}$ from Time-Series Data

In this section, we consider three 2-dimensional examples with varying geometry and two 6-dimensional physical systems. In the 2D experiments, we have dynamics evolve on a

manifold with boundary $M = [0,1]^2$ with linear dynamics $\dot{x} = Cx, \forall x \in [0,1]^2$.

We consider the reset map that gives the quotient manifold $M_\mathcal{H}$ a complicated topology by gluing two neighboring boundaries of $M$, i.e., $S = S_1 \cup S_2$ with $S_1 = \{x | x_2 \in [0,1], x_1 = 1\}$ and $S_2 = \{x | x_1 \in [0,1], x_2 = 1\}$ to their opposite sides. We consider the following reset function that makes $M_\mathcal{H}$ diffeomorphic to **Torus** and **Klein Bottle** by $r_\text{t}(x) = \begin{cases} (0, x_2) & x \in S_1 \\ (x_1, 0) & x \in S_2 \end{cases}$ and $r_\text{k}(x) = \begin{cases} (0, x_2) & x \in S_1 \\ (1 - x_1, 0) & x \in S_2 \end{cases}$.

The **Bouncing Ball** system satisfies the linear dynamics with gravity $g$ by $\dot{x}_1 = x_2, \dot{x}_2 = -g$. The guard surface $S_\text{b} := \{(x_1, x_2) | x_1 = 0, x_2 \leq 0\}$ indicates the ground where elastic collisions happen and is represented by the reset map $r_\text{b}(x) = [x_1, -\alpha x_2]^\top, [x_1, x_2]^\top \in S_\text{b}$.

The first 6-dimensional system is the **Three-Link Walker** governed by the rigid body dynamics in the continuous-time $D(q)\ddot{q} + C(q, \dot{q})\dot{q} + G(q) = \pi(q, \dot{q})$ with $q, \dot{q} \in \mathbb{R}^3$, $D(q)$ the mass matrix, $C(q, \dot{q})$ the Coriolis matrix, $G(q)$ the gravity vector and $\pi(q, \dot{q})$ an nonlinear feedback controller to enforce a periodic gait. The state reset happens when changing the stance and swing foot by the contact dynamics $\Delta$ (Featherstone, 2010) and the relabling of stance and swing foot by a permutation matrix $P$ (Westervelt et al., 2003) $\dot{q}^+ = \Delta(q^-, \dot{q}^+), q^+ = Pq^-$.

Then we consider a **3D-Bouncing Ball** thrown into a sink. The dynamics in the vertical direction are identical to the Bouncing Ball case, while the horizontal dynamics are not subject to any external force when in the air: $\dot{x} = v_x, \dot{v}_x = 0, \dot{y} = v_y, \dot{v}_y = 0$, but subject to the impact $r([x, v_x]^\top) = [x-, -\alpha v_x^-]^\top, \forall [x, v_x]^\top \in S_1 \cup S_2$, when reaching the wall defined by $S_\text{b1} := \{[x, v_x] : x = 0.8, v_x > 0\}$ and $S_\text{b2} := \{[x, v_x] : x = -0.8, v_x < 0\}$. The dynamics in $y$-direction are identical to $x$.

### 7.3. Comparison with Existing Methods

We compare Algorithm 1 with the existing methods for learning dynamical systems. The most relevant method

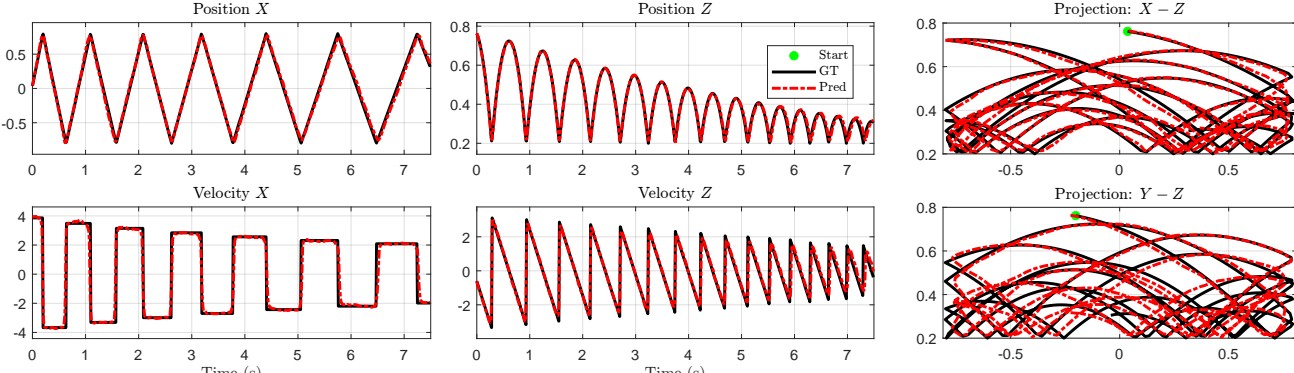

Figure 4: Trajectories of the 3D Bouncing Ball in a sink. The maximal training horizon in the curriculum is $2s$ while the testing set is $7.5s$. We see that the ball is colliding with the walls and the floor, but without obvious penetrations. The square wave in the horizontal velocity is extremely challenging for differentiable optimization that makes the baselines fail.

is (Teng et al., 2025) inspired by the Whitney Embedding Theorem to bias the latent space towards a continuous high-dimensional manifold by the gluing loss and latent consistency loss. We note that (Teng et al., 2025) admits a two-stage training strategy that learns the encoder and vector first and then the decoder. The second baseline is the classical Neural ODE (Chen et al., 2018) in the state space. The third baseline is the latent Neural ODE with RNN encoders (Rubanova et al., 2019). We also consider the deep Koopman operator (Lusch et al., 2018) that is designed for general dynamical systems. We note that all the baselines do not admit mode selections that are combinatorially hard.

We conduct the experiments five times for each example to compute the MSE (mean ± std.dev). We train the system for at most 200 steps in Neural ODE and rollout for more than 500 steps. For the Bouncing Ball, Klein Bottle, Torus, and Three-Link Walker, we consider the identical experimental settings in (Teng et al., 2025). For the 3D Bouncing Ball, we consider a larger neural network. For fairness, we consider that all activations in the hidden layers of the proposed method are ReLU, though in the ablations, we find that the sin activation function lead to better results. The details of the experiments are shown in Section C.1. The statistics of the experiments is shown in Table 1. We find that the proposed method greatly outperforms (Teng et al., 2025) in all the cases, while the other methods without the geometry-based inductive bias have different kinds of failures.

We plotted the trajectories predicted by Algorithm 1 of the 3D Bouncing Ball in Figure 4 and Figure 5. As the dynamics are decoupled in three directions, we present the trajectories in only $x-$ and $z-$directions. More visualizations are presented in Section C.2.

### 7.4. Ablation Studies

We conduct ablation studies to explore the key factors that contribute to the success of Algorithm 1. We denote the

Table 2: Ablation on the effect of the activation functions. We consider the activation function in the last layer as sin or ReLU.

| | sin, $\mathcal{L}_{x,z}$ | sin, $\mathcal{L}_{x,z,c}$ | ReLU, $\mathcal{L}_{x,z}$ | ReLU, $\mathcal{L}_{x,z,c}$ |
|---|---|---|---|---|
| Bouncing Ball | **0.106 ± 0.0164** | 0.117 ± 0.0326 | 0.253 ± 0.172 | 0.173 ± 0.0405 |
| Torus | 0.0163 ± 0.0305 | **0.00304 ± 0.00185** | 0.0411 ± 0.0400 | 0.00363 ± **0.000989** |
| Klein | 0.00954 ± 0.00807 | **0.00454 ± 0.00075** | 0.0499 ± 0.0263 | 0.0128 ± 0.0155 |
| Three-Link Walker | 0.0731 ± **0.00506** | 0.0888 ± 0.00794 | **0.0722 ± 0.00870** | 0.0963 ± 0.00685 |
| 3D Bouncing Ball | 0.175 ± 0.00949 | 0.169 ± 0.0106 | **0.157** ± 0.0111 | 0.162 ± 0.0150 |

loss function in Table 3 with different terms as $\mathcal{L}_{(\cdot,\cdot,\cdots)}$. We summarize the experiment setup in Section D.

**Activation functions:** As we see in the analytical examples, the glued space $M_{\mathcal{H}}$ may admit non-trivial topology that creates caveats that do not exhibit in the state space $M$, such as our analytical example. Thus, we design the first experiments to see if a $\sin(\cdot)$ activation function (Sitzmann et al., 2020) at the last hidden layer will improve the performance. We consider the loss function $\mathcal{L}_{x,z}$ or $\mathcal{L}_{x,z,c}$ with sin or ReLU activations. As shown in Table 2, we find that for the Torus or Klein Bottle case that has complex topologies, the gluing loss has significantly better performance then ReLU.

**Loss functions:** Then, we explore the dominant terms in the loss function (9) and summarize in Table 3. As shown in the last two columns, we find deactivating $\mathcal{L}_z$ results in significant performance degradation compared to its counterparts in the first and third columns. The fourth and fifth columns suggests that the $\mathcal{L}_v$ results in significant performance degradation for the Walker and 3D Bouncing Ball. For the first three columns, we conclude that activating $\mathcal{L}_g$ and the $\mathcal{L}_c$ simultaneously leads to better performance.

Therefore, we conclude that $\mathcal{L}_z$ is the key factor that ensures accurate reconstructions. As for the geometry-based loss, $\mathcal{L}_v$ does not help improve the performance, while $\mathcal{L}_c$ together $\mathcal{L}_g$ provides *marginal* improvements.

**Latent dimensions:** Finally, we analyze the effect of the latent dimensions. We consider the $m = n + k$, $m = 2n$, and $m = 2n + 2$, where $k < n$ depends on the knowledge

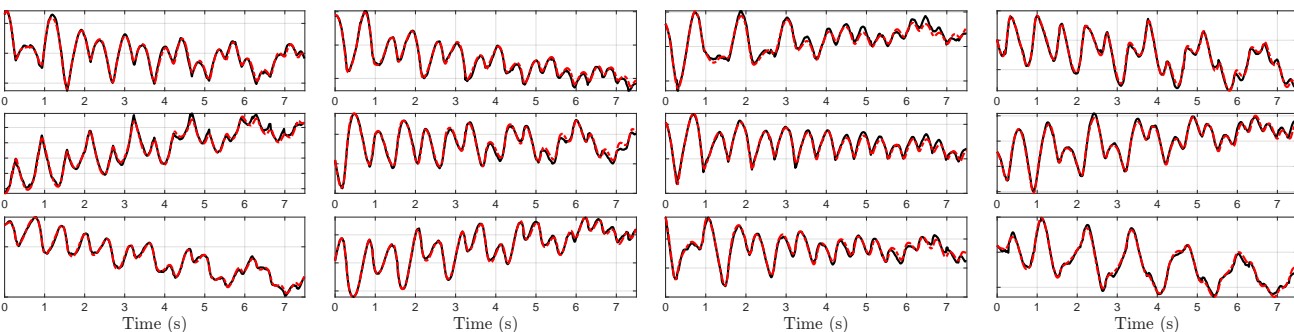

Figure 5: The continuous latent trajectories ($m = 12$) of the 3D Bouncing Ball.

Table 3: Ablation study on the effect on combination of loss function $\mathcal{L}_{(\cdot)}$ on the reconstruction MSE (mean ± std.dev). We find that $\mathcal{L}_v$ does not improve the performance. Without latent consistency loss $\mathcal{L}_z$, the performance degrades a lot. $\mathcal{L}_g$ and $\mathcal{L}_c$ together have an improvement. The first three columns do not show a significant difference, while $\mathcal{L}_z$ is the dominant factor in the success.

| | $\mathcal{L}_{x,z}$ | $\mathcal{L}_{x,z,c}$ | $\mathcal{L}_{x,z,c,g}$ | $\mathcal{L}_{x,z,g,v}$ | $\mathcal{L}_{x,z,c,g,v}$ | $\mathcal{L}_x$ | $\mathcal{L}_{x,c,g}$ |
|---|---|---|---|---|---|---|---|
| Bouncing Ball | 0.106 ± **0.0164** | 0.117 ± 0.0330 | **0.0996** ± 0.0255 | 0.118 ± 0.0251 | 0.122 ± 0.0506 | 0.738 ± 0.118 | 0.945 ± 0.518 |
| Torus | 0.0163 ± 0.0305 | 0.00304 ± 0.00185 | **0.00280** ± 0.00065 | 0.0196 ± 0.0276 | 0.00336 ± 0.00080 | 0.0742 ± 0.0131 | 0.105 ± 0.0799 |
| Klein | 0.00954 ± 0.00807 | **0.00454** ± 0.000750 | 0.00464 ± **0.00074** | 0.0380 ± 0.0315 | 0.00516 ± 0.00106 | 0.0737 ± 0.0111 | 0.0706 ± 0.0145 |
| Three-Link Walker | **0.0731** ± **0.00506** | 0.0888 ± 0.00794 | 0.0913 ± 0.00944 | 0.118 ± 0.00774 | 0.131 ± 0.0221 | 0.236 ± 0.161 | 0.126 ± 0.0115 |
| 3D Bouncing Ball | 0.175 ± **0.00949** | 0.169 ± 0.0106 | **0.160** ± 0.0205 | 0.254 ± 0.0271 | 0.197 ± 0.0173 | 0.416 ± 0.0905 | 0.455 ± 0.101 |

Table 4: Ablation on the effect of the latent dimensions on the reconstruction MSE (mean ± std.dev).

| | $m = n + k$ | $m = 2n$ | $m = 2n + 2$ |
|---|---|---|---|
| Bouncing Ball | 0.114 ± **0.0243** | **0.0996** ± 0.0255 | 0.119 ± 0.0275 |
| Torus | 0.00537 ± 0.00354 | 0.00280 ± 0.00065 | **0.00216** ± **0.000187** |
| Klein | 0.0194 ± 0.00595 | 0.00464 ± **0.00074** | **0.00457** ± 0.00079 |
| Three-Link Walker | 0.110 ± **0.00499** | 0.0913 ± 0.00944 | **0.0891** ± 0.00841 |
| 3D Bouncing Ball | 0.481 ± 0.145 | **0.160** ± 0.0205 | 0.167 ± **0.0137** |

of the underlying system topology. For the Bouncing Ball and the 3D version, we consider $k = 0$ as their dynamics in each DOF can be embedded into flows on a 2D plane (Simic et al., 2005). For the Torus, we consider $k = 1$ as a Torus can be embedded into a $\mathbb{R}^3$. For the Klein Bottle, we also consider $k = 1$ though it cannot be embedded into $\mathbb{R}^3$.

As illustrated in Table 4, we find that lowering the dimensions leads to significant performance degradation, while increasing the dimensions generally leads to improved performance. For the 1D Bouncing Ball case, we note that lower-dimension does not lead to significant performance degradation, as the underlying system is can be embedded into a two-dimensional space. For the Klein Bottle, as the underlying manifold can not be embedded into $\mathbb{R}^3$, the performance degradation is significant.

## 8. Discussions and Limitations

**Why generic property:** Our proof shows that satisfying (C-1) and (C-2) is not a rare case, but a property that holds for the majority of the admissible constructions. Thus, learning these functions does not fit a singular case, but searching within a broad class of well-behaved embeddings. This can be done by simply letting $m > 2n$.

**Dimension bound:** In this work, we proved the dimension bound $m > 2n$ through the transversality theorem. For $m = 2n$, the Theorem 3 and 4 suggest the singular case has dimension 0, i.e., countably or finitely many (compact state space) discrete points. To explore Theorem 6 when $m = 2n$, we need to eliminate these discrete points with additional techniques (Hirsch, 2012).

**Corner of the guard surface:** The main proof assumes the guard surface is smooth. In case that $S$ is piecewise smooth, we need to discuss the transversality condition on the intersection of different pieces. As the corners may have degeneracies (local linear dependencies) that require careful discussion, we consider this an interesting future extension.

**Significance of loss design:** The ablation study indicates that the $\mathcal{L}_z$ plays a dominant role in performance, while $\mathcal{L}_v$ does not yield additional gains, and $\mathcal{L}_c$ and $\mathcal{L}_g$ provide only marginal improvements. Importantly, this observation does not contradict our theoretical formulation. The velocity compatibility. (C-2) can be implicitly enforced by the finite Lipschitz continuity of the neural vector field, a property commonly satisfied by standard network parameterizations without requiring an explicit loss term (Virmaux & Scaman, 2018). As a result, $\mathcal{L}_v$ can be redundant in practice.

**Discontinuity of the decoder:** We note that the embedding $f$ glues two distinct points in the state space as a single point in the latent space. Like in (Teng et al., 2025), we leave the discontinuity to the decoder. To encourage a strict discontinuous decoder, we may adopt the atlas learning that applies a single network to each chart to make the decoder truly discontinuous, or adopt step functions in the decoder networks to recover a sharper discontinuity.

# 9. Conclusions

In this work, we theoretically proved that an $n-$dimensional hybrid system can be embedded into $m$-dimensional Euclidean space with a continuous vector field in its embedded image whenever $m > 2n$. Building on this existence theorem, we show that the proposed latent Neural ODE is well-posed to learn hybrid systems from time-series data with superior accuracy. The ablation study suggested that the consistency loss in both the state and the latent space is the key to successful implementations.

# Acknowledgment

This work is supported in part by The Robotics and AI Institute.

# Impact Statement

This paper presents work whose goal is to advance the field of machine learning. There are many potential societal consequences of our work, none of which we feel must be specifically highlighted here.

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

# A. Topology of Function Space

## A.1. Differential Geometry

**Definition 5** (Chart and Atlas (Hirsch, 2012)). *Let $M$ be a topological $n$-manifold. A* chart *on $M$ is a pair $(U, \phi)$, where $U \subseteq M$ is an open set and $\phi : U \to \hat{U} \subseteq \mathbb{R}^n$ is a homeomorphism onto an open subset $\hat{U}$ of $\mathbb{R}^n$. An* atlas $\mathcal{A}$ *for $M$ is a collection of charts $\{(U_\alpha, \phi_\alpha)\}_{\alpha \in \mathcal{A}}$ whose domains cover $M$, i.e., $M = \bigcup_{\alpha \in \mathcal{A}} U_\alpha$.*

*Two charts $(U, \phi)$ and $(V, \psi)$ are said to be* smoothly compatible *if either $U \cap V = \emptyset$, or the transition map $\psi \circ \phi^{-1} : \phi(U \cap V) \to \psi(U \cap V)$ is a diffeomorphism between open subsets of $\mathbb{R}^n$. An atlas $\mathcal{A}$ is called a* smooth atlas *if any two charts in $\mathcal{A}$ are smoothly compatible.*

**Definition 6** (Bump function (Hirsch, 2012)). *Let $M$ be a smooth manifold. A* bump function *on $M$ is a smooth function $\rho : M \to \mathbb{R}$ with compact support, i.e. $\rho \in C_c^\infty(M)$.*

*In particular, on $\mathbb{R}^n$ one may construct a bump function as follows. Define*

$$\eta(t) = \begin{cases} \exp\left(-\frac{1}{1-t}\right), & t < 1, \\ 0, & t \geq 1, \end{cases} \qquad \psi(t) = \frac{\eta(t)}{\eta(t) + \eta(1-t)}. \tag{14}$$

*Then $\psi \in C^\infty(\mathbb{R})$ satisfies $\psi(t) = 1$ for $t \leq 0$ and $\psi(t) = 0$ for $t \geq 1$. Setting $\varphi(x) = \psi(\|x\|^2)$ yields a smooth function $\rho \in C_c^\infty(\mathbb{R}^n)$ supported in the unit ball and identically equal to $1$ in a neighborhood of the origin. Via local coordinate charts, such bump functions exist on any smooth manifold.*

## A.2. The Space of $C^k$ Maps

**Definition 7** (Whitney $C^k$ topology (Abraham & Robbin, 1967)). *Let $M$ be a compact smooth manifold and $N$ a smooth manifold. Fix a finite atlas $\{(U_i, \phi_i)\}$ of $M$ and charts $\{(V_i, \psi_i)\}$ of $N$ such that $f(U_i) \subset V_i$. For $\epsilon > 0$, define the basic neighborhood $B_\epsilon(f)$ to be the set of all $g \in C^k(M, N)$ satisfying $g(U_i) \subset V_i$ and*

$$\left\| D^\alpha(\psi_i \circ g \circ \phi_i^{-1})(\phi_i(x)) - D^\alpha(\psi_i \circ f \circ \phi_i^{-1})(\phi_i(x)) \right\| < \epsilon$$

*for all $i$, all $x \in U_i$, and all multi-indices $|\alpha| \leq k$. The Whitney $C^k$ topology on $C^k(M, N)$ is the topology generated by these neighborhoods. When $N = \mathbb{R}^m$, this topology is induced by the norm*

$$\|f\|_{C^k} := \max_{|\alpha| \leq k} \sup_{x \in M} \|D^\alpha f(x)\|,$$

*hence $C^k(M, \mathbb{R}^m)$ is a Banach space. For general $N$, $C^k(M, N)$ carries a natural structure of a Banach manifold. Here $\|\cdot\|$ denotes any norm on the relevant finite-dimensional vector space; different choices yield the same topology.*

## A.3. Banach Manifolds

**Definition 8** (Banach manifold (Abraham & Robbin, 1967)). *A Banach manifold $\mathcal{P}$ is a Hausdorff, second-countable topological space equipped with an atlas $\{(U_i, \phi_i)\}$ such that $\{U_i\}$ is an open cover of $\mathcal{P}$, each chart $\phi_i : U_i \to E_i$ is a homeomorphism onto an open subset of a Banach space $E_i$, and for any indices $i, j$ with $U_i \cap U_j \neq \varnothing$, the transition map $\phi_j \circ \phi_i^{-1} : \phi_i(U_i \cap U_j) \to \phi_j(U_i \cap U_j)$ is $C^k$ in the sense of Fréchet differentiability.*

**Remark 4.** *The parameter space $\mathcal{P}$ will typically be taken to be a $C^k$ Banach manifold. In particular, if $M$ is compact, the mapping space $C^k(M, N)$ admits a natural $C^\infty$ Banach manifold structure constructed using local charts of $N$ and the $C^k$ topology on $M$. When $N = \mathbb{R}^m$, this structure is modeled on the Banach space $C^k(M, \mathbb{R}^m)$ with norm*

$$\|f\|_{C^k} := \max_{|\alpha| \leq k} \sup_{x \in M} \|D^\alpha f(x)\|.$$

*For general target manifolds $N$, the resulting Banach manifold structure is canonical up to smooth equivalence. This framework allows transversality results to be applied not only to finite-dimensional parameter families but also to infinite-dimensional spaces of perturbations of a given map; see (Abraham & Robbin, 1967).*

## A.4. Baire Space and Residual Sets

**Definition 9** (Baire Space (Abraham & Robbin, 1967)). *A topological space $X$ is a Baire space if for every countable collection of open dense subsets $\{\mathcal{V}_n\}_{n=1}^{\infty}$ in $X$, their intersection $\bigcap_{n=1}^{\infty} \mathcal{V}_n$ is dense in $X$.*

**Definition 10** (Residual Set (Abraham & Robbin, 1967)). *A subset $R \subseteq X$ is called a residual set (or a generic set) if it contains a countable intersection of open dense sets. By the Baire Category Theorem, every complete metric space (and thus every Banach manifold) is a Baire space. In this paper, we say a property of the embedding is generic if the set of functions $h \in \mathcal{P}$ satisfying the property is a residual set in the Whitney $C^k$ topology.*

**Example 1.** *Let $X = \mathbb{R}$ with the standard Euclidean topology. We consider the following subsets to illustrate the hierarchy of density and genericity:*

- ***Dense set:*** *The set of rational numbers $\mathbb{Q}$ is dense in $\mathbb{R}$ because every non-empty open interval $(a, b)$ contains a rational number. However, it is not a Baire space in its relative topology, nor is it open or residual.*

- ***Open dense set:*** *Let $S = \mathbb{R} \setminus \mathbb{Z}$ (the reals minus the integers). This set is open (the union of intervals $(n, n+1)$) and dense in $\mathbb{R}$. The open and dense set is also residual.*

- ***Residual set (Generic set):*** *The set of irrational numbers $\mathbb{R} \setminus \mathbb{Q}$ is a residual set. Since $\mathbb{Q} = \{q_1, q_2, \dots\}$ is countable, we can write $\mathbb{R} \setminus \mathbb{Q} = \bigcap_{n=1}^{\infty}(\mathbb{R} \setminus \{q_n\})$. Each $V_n = \mathbb{R} \setminus \{q_n\}$ is open and dense, making their intersection residual. Note that while $\mathbb{Q}$ and $\mathbb{R} \setminus \mathbb{Q}$ are both dense, only the latter is residual.*

- ***Nowhere dense set:*** *The standard Cantor set $\mathcal{C}$ or the set of integers $\mathbb{Z}$ are nowhere dense. Their closures have an empty interior; they contain no open intervals, meaning they are "full of holes" at every scale.*

**Remark 5** (Countable intersection). *The countable intersection of a residual set is also residual. Thus, we can leverage this property to find residual sets that satisfy different conditions, and their non-empty intersection is residual and satisfies all the conditions.*

## A.5. Transversality Conditions

Though the transversality condition usually requires checking both the differential and the tangent tangent space of $Z$, we note that if the tangent map spans the entire space of the image, the transversality condition is satisfied automatically:

**Proposition 1** (Sufficient Condition for Transversality). *Let $\Gamma : M \times P \to N$ be the total map as defined in Theorem 4. The transversality condition $\Gamma \pitchfork Z$ is satisfied if, for every $(x, p) \in \Gamma^{-1}(Z)$, the total differential*

$$d\Gamma_{(x,p)} : T_x M \times T_p P \to T_{\Gamma(x,p)} N \tag{15}$$

*is surjective.*

*Proof.* By definition, the map $\Gamma$ is transversal to $Z$ if the following condition holds for all $(x, p) \in \Gamma^{-1}(Z)$:

$$\mathrm{Im}(d\Gamma_{(x,p)}) + T_{\Gamma(x,p)}Z = T_{\Gamma(x,p)}N. \tag{16}$$

If the total differential $d\Gamma_{(x,p)}$ is surjective, its image is the entire tangent space, i.e., $\mathrm{Im}(d\Gamma_{(x,p)}) = T_{\Gamma(x,p)}N$. Consequently, the sum of $\mathrm{Im}(d\Gamma_{(x,p)})$ and any subspace of $T_{\Gamma(x,p)}N$ (including $T_{\Gamma(x,p)}Z$) trivially recovers the entire tangent space $T_{\Gamma(x,p)}N$.

$\square$

# B. Proof of Theorem 6

Consider the main existence theorem of this work:

**Theorem 6** (Continuous Extrinsic Representation of Hybrid Systems). *For a hybrid system $\mathcal{H}$ satisfying Assumption 1, 2 and 3, whenever $m > 2n$, there exists an $f \in C^k(M, \mathbb{R}^m)$ that satisfies:*

$$f(x) = f(r(x)), \forall x \in S \tag{C-1}$$

$$Df(x)V(x) = Df(r(x))V(r(x)), \forall x \in S \tag{C-2}$$

$$f \text{ induces an embedding of } M_{\mathcal{H}} \text{ into } \mathbb{R}^m. \tag{C-3}$$

*Moreover, $f$ can be obtained by a generic choice within the admissible construction space.*

We apply four steps to show the existence of $f$: (1) Design $f$ on the guard surface $S$ and its image after the reset $r(S)$; (2) Decide when $f$ is injective and surjective; and (3) Extend the property from the neighborhood of $S \cup r(S)$ to entire $M$. The geometry of the construction is illustrated in Figure 2.

For simplification, we denote $R := r(S)$ as the image of $r(x), \forall x \in S$. In this work, we use the Whitney Topology as shown in Definition 7 for all $C^k$ functions in the Baire space.

## B.1. Determine the Embedding on the Guard Surface

In this section, we show that we can enforce (C-1) by an embedding $g : S \to \mathbb{R}^m$ to specify the value on $S$ and enforce (C-2) by $g_S : S \to \mathbb{R}^m$ to determine the derivative and value inward $S$.

**Lemma 1** (Value compatibility). *Consider the embedding on the boundary: $f_S : S \to \mathbb{R}^m$ and $f_R : R \to \mathbb{R}^m$. When $m > 2(n-1)$, there exist a smooth embedding $g : S \to \mathbb{R}^m$ such that $f_S(x) = g(x)$ and $f_R(x) = g \circ r^{-1}(x)$ that satisfy $f_S(x) = f_R(r(x)), \forall x \in S$.*

*Proof.* Since $S$ is a smooth $(n-1)$-dimensional embedded submanifold, Theorem 5 ensures the set of a smooth embedding $g : S \to \mathbb{R}^m$ is residual (Hirsch, 2012; Abraham & Robbin, 1967) when $m > 2(n-1)$. Then we choose such a $g$ to determine the value of $f$ on $S \cup R$:

$$f_S(x) = g(x), \forall x \in S, \tag{17}$$

$$f_R(y) = g \circ r^{-1}(y), \forall y \in R. \tag{18}$$

Then for every $x \in S$ we can verify that (C-1) is satisfied on $S \cup R$ by checking:

$$f_R(r(x)) = g \circ r^{-1}(r(x)) = g(x) = f_S(x). \tag{19}$$

$\square$

Given $g$, we can enforce (C-1) of $f$ on the $S$- and $R$-side in the collar coordinates. Then, we introduce the first-order collar extension to extend the value of $f$ into the interior of $M$.

**Remark 6** (First-order collar extension). *Given boundary embedding $f_S : S \to \mathbb{R}^m$, the collar embedding $\kappa_S : S \times [0, \epsilon) \to M$, and a prescribed smooth function $g_S : S \to \mathbb{R}^m$, the map $f_S(x, t) : \kappa_S \to M$*

$$\bar{f}_S(x, t) = f_S(x) + t\, g_S(x) \tag{20}$$

*defines the first-order Taylor extension of $f$ along the collar direction $t$. This construction enforces the boundary value and the derivative inward $\partial M$ at $t = 0$ while leaving higher-order behavior unspecified.*

Then we show that once the derivative on the $S$-side is determined by the first-order extension $\bar{f}_S$, the collar extension on $R$-side, i.e., $\bar{f}_R$ is uniquely determined if (C-2) is enforced.

**Lemma 2** (Derivative compatibility). *Under Assumption 2, for any smooth function $g_S : S \to \mathbb{R}^m$, there exists a unique $g_R : R \to \mathbb{R}^m$ such that the first-order collar extension of $\bar{f}_S(x)$ and $\bar{f}_R(x)$ satisfy the (C-2) for $\forall x \in S$.*

*Proof.* As $S \in \partial M$ is an $(n-1)$-dimensional submanifold on the boundary, we study the property of $f$ on $S \cup R$ on the collar coordinate for convenience.

With an sufficiently small $\epsilon > 0$, we define the collar embedding on $S$ and $R$ as

$$\kappa_S : S \times [0, \epsilon) \to M \text{ and } \kappa_R : R \times [0, \epsilon) \to M.$$

Thus, we have $\kappa_R$ is an immersion and we can define the pullbacks of $V$ in collar coordinates:

$$\overline{V}_S(x, t) = (D\kappa_S(x, t))^{-1} V(\kappa_S(x, t)) \tag{21}$$

By the direct sum decomposition of $T_x M, x \in S$, into the tangential component on $T_x S$ and the direction into the interior of $M$, i.e., $T_{(x,0)}(S \times [0, \epsilon)) \cong T_x S \oplus \text{span}\{\partial_t\}$, we decompose the vector field $\overline{V}_{(\cdot)}$ uniquely by:

$$\overline{V}_S(x, 0) = \xi_S(x) + \nu_S(x)\partial_t, \tag{22}$$

with $\xi_S \in T_x S$ and $\nu_S \in \mathbb{R}$ (and similarly on $R$).

Choose a $g_S : S \to \mathbb{R}^m$ and $g_R : R \to \mathbb{R}^m$, we construct the first-order collar extensions of $f_S$ and $f_R$:

$$\begin{aligned}
\bar{f}_S(x, t) &= f_S(x) + tg_S(x), \forall x \in S, \\
\bar{f}_R(y, t) &= f_R(y) + tg_R(y), \forall y \in R.
\end{aligned} \tag{23}$$

We can see that (23) satisfy the boundary value conditions by checking that whenever $y = r(x)$:

$$\bar{f}_S(x, 0) = f_S(x) = (f_S \circ r^{-1}) \circ r(x) = f_R(r(x)) = \bar{f}_R(r(x), 0) \tag{24}$$

Morever, the prescribed derivative into the interior of $S$ and $R$ can be determined by

$$\partial_t \bar{f}_S(x, 0) = g_S(x), \partial_t \bar{f}_R(x, 0) = g_R(x). \tag{25}$$

By the direct sum decomposition of the tangent map on the collar coordinate and the decomposition of $\overline{V}_S$ in (22) for $\xi_S \in T_x S$ and $\nu_S \in \mathbb{R}$, we have the unique decomposition:

$$D\bar{f}_S(x, 0)\overline{V}_S(x, 0) = Df_S(x)\xi_S(x) + \nu_S(x)g_S(x), \tag{26}$$

and similarly for the $R$-side:

$$D\bar{f}_R(r(x), 0)\overline{V}_R(r(x), 0) = Df_R(r(x))\xi_R(r(x)) + \nu_R(r(x))g_R(r(x)). \tag{27}$$

To impose (C-2) for $\forall x \in S$, we enforce the require (26) and (27) to equal:

$$D\bar{f}_S(x, 0)\overline{V}_S(x, 0) = D\bar{f}_R(r(x), 0)\overline{V}_R(r(x), 0). \tag{28}$$

Thus, we yield the equation to decide $g_R$ from $g_S$

$$\nu_R(r(x))g_R(r(x)) = Df_S(x)\xi_S(x) + \nu_S(x)g_S(x) - Df_R(r(x))\xi_R(r(x)). \tag{29}$$

Assumption 2 implies the flow transverses $S$ by pointing outward and transverse $R$ by pointing inward, hence

$$\nu_S(x) < 0, \forall x \in S, \text{ and } \nu_R(y) > 0, \forall y \in R. \tag{30}$$

Thus, solving (29) yield a *unique* solution

$$g_R(r(x)) = \frac{\nu_S(x)}{\nu_R(r(x))} g_S(x) + c(x), \tag{31}$$

with a fixed smooth term $c(x)$ independent of the choice of $g_S$:

$$c(x) = \frac{Df_S(x)\xi_S(x) - Df_R(r(x))\xi_R(r(x))}{\nu_R(r(x))}. \tag{32}$$

$\square$

All terms on the RHS of (31) depend smoothly on $x$, so $g_R$ is smooth on $R$ and uniquely determined by $g_S$. Thus, we can control $g_S$ to enforce (C-2) on $S \cup R$.

## B.2. Immersion and Injectivity on the Collar Neighborhood

### B.2.1. IMMERSION ON COLLAR NEIGHBORHOOD

Now we derive the condition on which the first-order collar extension (20) is an immersion (full rank) in the neighborhood of $S \cup U$. Define collar neighborhood $U := U_S \cup U_R$ with

$$U_S = \kappa_S(S \times [0, \epsilon)), U_R = \kappa_R(R \times [0, \epsilon)). \tag{33}$$

For fixed $g$ and $g_S$, by Lemma 1 and 2, we have the candidate $f$ defined on $U$ induced by the collar embedding:

$$f : U \to \mathbb{R}^m, \quad f(x) = \begin{cases} \bar{f}_S(\kappa_S^{-1}(x)), x \in U_S \\ \bar{f}_R(\kappa_R^{-1}(x)), x \in U_R \end{cases}. \tag{34}$$

Now we proceed to prove that $f$ is an immersion on $U$ for a generic choice of $g_S$.

**Lemma 3** (Generic immersion on the collar). *Assume $m > 2n - 2$. For a fixed embedding $g : S \to \mathbb{R}^m$, there exists a residual subset $\mathcal{F}_{\mathrm{imm}} \subset C^k(S, \mathbb{R}^m)$ such that for all $g_S \in \mathcal{F}_{\mathrm{imm}}$, the map $f$ defined by (34) is an immersion on $U$.*

*Proof.* Recall the first-order collar extension (23), where $f_S = g$ is a smooth embedding of the $(n-1)$-manifold $S$ and $f_R = g \circ r^{-1}$, thus we have $Df_S : T_x S \to \mathbb{R}^m$ and $Df_R : T_x R \to \mathbb{R}^m$ to satisfy the following rank condition:

$$\mathrm{rank}(Df_S(x)) = \mathrm{rank}(Df_R(y)) = n - 1. \tag{35}$$

Therefore $D\bar{f}_S(x, 0)$ and $D\bar{f}_R(y, 0)$ have rank $n$ if and only if $g_S(x)$ and $g_R(y)$ satisfy

$$g_S(x) \notin \mathrm{Im}(Df_S(x)), \tag{36}$$

$$g_R(y) \notin \mathrm{Im}(Df_R(y)) \Rightarrow g_S(x) \notin A_S(x). \tag{37}$$

with $A_S(x) \subset \mathbb{R}^m$ the affine subspace obtained by pulling back $\mathrm{Im}(Df_R(r(x)))$ under (31). We note that both $\mathrm{Im}(Df_S)$ and $A_S(x)$ are independent of $g_S$.

For fixed $g$, define the subsets of $S \times \mathbb{R}^m$ that violate (36) and (37):

$$\begin{aligned} B_S &:= \{(x, v) : v \in \mathrm{Im}(Df_S(x))\}, \\ B_R &:= \{(x, v) : v \in A_S(x)\}. \end{aligned} \tag{38}$$

Since $Df_S$ has constant rank $n - 1$, its image $\{\mathrm{Im}(Df_S(x))\}_{x \in S}$ forms a smooth vector subbundle $E_S$ of the trivial bundle $S \times \mathbb{R}^m$. By the constant rank theorem, the family of subspaces $\{\mathrm{Im}(Df_S(x))\}_{x \in S}$ fits together to form a smooth vector subbundle. The total space $B_S = \{(x, v) \in S \times \mathbb{R}^m : v \in \mathrm{Im}(Df_S(x))\}$ is therefore an embedded submanifold. Similarly, for $B_R$, as each $A_S(x)$ is obtained via a smooth affine pullback of the constant-rank image of $Df_R$, the collection $\{A_S(x)\}_{x \in S}$ defines a smooth affine subbundle. Its total space $B_R$ is likewise an embedded submanifold of $S \times \mathbb{R}^m$.

Thereofore, both $B_S$ and $B_R$ have dimension

$$\dim B_S = \dim B_R = (n - 1) + (n - 1), \tag{39}$$

while the ambient space $S \times \mathbb{R}^m$ has dimension $(n - 1) + m$. Hence we have

$$\mathrm{codim}(B_S) = \mathrm{codim}(B_R) = m - (n - 1). \tag{40}$$

Then we apply Theorem 4. We first define the following total map:

$$\Gamma : S \times C^k(S, \mathbb{R}^m) \to S \times \mathbb{R}^m, (x, g) \mapsto (x, g(x)). \tag{41}$$

For fixed $g_S \in C^k(S, \mathbb{R}^m)$ we have the slice map:

$$\Gamma(\cdot \mid g_S) : S \to S \times \mathbb{R}^m, x \to (x, g_S(x)). \tag{42}$$

We claim that $\Gamma$ is a submersion, which implies it is transverse to any submanifold of $S \times \mathbb{R}^m$, including $B_S$ and $B_R$. At any point $(x, g)$, the differential $d\Gamma_{(x,g)} : T_x S \times C^k(S, \mathbb{R}^m) \to T_x S \times \mathbb{R}^m$ is given by:

$$d\Gamma_{(x,g)}(u, h) = \big(u,\ Dg(x)u + h(x)\big), \tag{43}$$

with $u \in T_x S$ and $h \in C^k(S, \mathbb{R}^m)$. To show surjectivity of $\Gamma$, let $(u, w) \in T_x S \times \mathbb{R}^m$ be an arbitrary vector. By choosing $h(y) = \rho(y)(w - Dg(x)u)$, where $\rho \in C_c^\infty(S)$ is a local bump function supported near $x$ with $\rho(x) = 1$. Thus, we can see that $h(x)$ can reach any value specified by $u$, and thus $d\Gamma_{(x,g)}$ is surjective at every point. Therefore $\Gamma \pitchfork B_S$ and $\Gamma \pitchfork B_R$ on $S \times C^k(S, \mathbb{R}^m)$. By Theorem 4, the sets

$$\mathcal{F}_1 = \{g_S \in C^k(S, \mathbb{R}^m) :\ \Gamma(\cdot \mid g_S) \pitchfork B_S\} \text{ and} \tag{44}$$

$$\mathcal{F}_2 = \{g_S \in C^k(S, \mathbb{R}^m) :\ \Gamma(\cdot \mid g_S) \pitchfork B_R\}, \tag{45}$$

are residual in $C^k(S, \mathbb{R}^m)$, so as their intersection

$$\mathcal{F}_{\text{imm}} := \mathcal{F}_1 \cap \mathcal{F}_2. \tag{46}$$

Thus, for $\forall g_S \in \mathcal{F}_{\text{imm}}$, $\Gamma(\cdot \mid g_S) \pitchfork B_S$ and $\Gamma(\cdot \mid g_S) \pitchfork B_R$. By Theorem 3, $\Gamma^{-1}(B_S \mid g_S)$ is a submanifold of $S$ with the dimension of the preimage being

$$\dim \Gamma^{-1}(B_S \mid g_S) = \dim S - \text{codim}(B_S), \tag{47}$$

By $\dim S = n - 1$ and (40) for $\text{codim}(B_S) = m - (n - 1)$, we have

$$\dim \Gamma^{-1}(B_S \mid g_S) = (n - 1) - (m - (n - 1)) = 2(n - 1) - m. \tag{48}$$

If $m > 2n - 2$, the dimension of the preimage is negative, hence $\Gamma^{-1}(B_S \mid g_S) = \emptyset$. Similarly, $m > 2n - 2 \Rightarrow \Gamma^{-1}(B_R \mid g_S) = \emptyset$. Equivalently, (36) and (37) are satisfied for $\forall x \in S$ when $m > 2n - 2$, which ensures a generic choice of $g_S \in \mathcal{F}_{\text{imm}}$ makes $f$ has full rank when evaluating on $S$.

Since having full rank is an open condition in the space of linear maps, and since $(x, t) \mapsto D\bar{f}_S(x, t)$ is continuous, the fact that $D\bar{f}_S(x, 0)$ has full rank for all $x \in S$ implies that for each $x \in S$ there exist a neighborhood $U_x \subset S$ and $\epsilon_x > 0$ such that $D\bar{f}_S$ has full rank on $U_x \times [0, \epsilon_x)$. By the compactness of $S$, we may extract a finite subcover and obtain a uniform $\epsilon > 0$ such that $D\bar{f}_S$ has full rank on $S \times [0, \epsilon)$. $\qquad\square$

### B.2.2. INJECTIVITY ON THE COLLAR NEIGHBORHOOD

We derive the condition on which $x \neq y \Rightarrow f(x) \neq f(y), \forall x, y \in U/\sim$.

**Lemma 4** (Injectivity on the collar). *Assume $m > 2n$. For a fixed boundary embedding $g$, there exists a residual subset $\mathcal{F}_{\text{inj}} \subset C^k(S, \mathbb{R}^m)$ such that for all $g_S \in \mathcal{F}_{\text{inj}}$, the map $f$ is injective on $U/\sim$.*

*Proof.* As $f$ is defined on $U = U_S \cup U_R$ that has $S$ and $R$ sides, we split the proof to three conditions (1) $x, y \in U_S$, (2) $x, y \in U_R$, and (3) $x \in U_S, y \in U_R$.

***Injectivity on $S-$ or $R-$ Side:***

Consider $\Delta_S := \{(a, b) \mid a = b, (a, b) \in U_S \times U_S\}$ and we define $\Omega_S := \big(U_S \times U_S\big) \setminus \Delta_S$, so $(x, t) \neq (y, s)$ for every $((x, t), (y, s)) \in \Omega_S$. Then we consider the total map:

$$\Gamma_S : \Omega_S \times C^k(S, \mathbb{R}^m) \to \mathbb{R}^m \tag{49}$$

defined by the difference of $\bar{f}_S$ between two points

$$\Gamma_S(((x, t), (y, s)), g_S) := \big(f_S(x) + t g_S(x)\big) - \big(f_S(y) + s g_S(y)\big). \tag{50}$$

If $t = s = 0$, then $\Gamma_S = f_S(x) - f_S(y) = 0$, if and only if $x = y$, as $f_S$ is chosen to be an embedding.

Then we consider the condition $(t, s) \neq (0, 0)$. We derive the differential of $\Gamma_S$ in the parameter direction:

$$D_{g_S}\Gamma_S[h] = t\,h(x) - s\,h(y), h \in C^k(S, \mathbb{R}^m). \tag{51}$$

Similar to the proof of surjectivity for (43), we can choose the bump function to ensure the linear map $h \mapsto t\,h(x) - s\,h(y)$ can achive any value on $\mathbb{R}^m$ whenever $(t, s) \neq (0, 0)$. Thus, we can make the total map $\Gamma_S$ surjective everywhere by varying the parametric direction, which ensures the condition in Proposition 1 is satisfied. Hence $\Gamma_S$ is transverse to any manifold and particularly $\Gamma_S \pitchfork \{0\}$.

By Theorem 4, $\Gamma_S \pitchfork \{0\}$ ensures

$$\mathcal{F}_3 = \{g_S \in C^k(S, \mathbb{R}^m) : \Gamma_S(\cdot \mid g_S) \pitchfork \{0\}\}, \tag{52}$$

is residual. By Theorem 3, $\Gamma_S^{-1}(0 \mid g_S)$ is of dimension

$$\dim \Gamma_S^{-1}(0 \mid g_S) = \dim \Omega_S - m = 2n - m. \tag{53}$$

If $m > 2n$, this dimension is negative, hence the $\Gamma_S^{-1}(0 \mid g_S)$ is empty. Equivalently, for $\forall g_S \in \mathcal{F}_3$, $f$ is injective on $\Omega_S$.

Similarly, consider $\Omega_R := (U_R \times U_R) \setminus \Delta_R$ with $\Delta_R := \{(a, b)|a = b, (a, b) \in U_R \times U_R\}$. For $y_1, y_2 \in R, t, s \in [0, \epsilon)$ define the total map

$$\Gamma_R(((y_1, t), (y_2, s)), g_S) := (f_R(y_1) + t\,g_R(y_1)) - (f_R(y_2) + s\,g_R(y_2)) \tag{54}$$

By (31), we have $g_R$ depends smoothly on $g_S$. By parameterizing $y_1 = r(x_1)$ and $y_2 = r(x_2)$ as $r$ is an diffeomorphism, we convert (54) to:

$$\Gamma_R(((y_1, t), (y_2, s)), g_S) = \left(f_R(r(x_1)) + \frac{t\,\nu_S(x_1)}{\nu_R(r(x_1))}\,g_S(x_1) + tc(x_1)\right) - \left(f_R(r(x_2)) + \frac{s\,\nu_S(x_2)}{\nu_R(r(x_2))}\,g_S(x_2) + s\,c(x_2)\right) \tag{55}$$

We note that by Assumption 2, we have $\frac{\nu_S(x)}{\nu_R(r(x))} < 0$. Thus when $(t, s) \neq (0, 0)$, the value is dependent on $g_S$.

Similar to the proof in the $S$-side, one checks that the total map $\Gamma_R : \Omega_R \times C^k(S, \mathbb{R}^m) \to \mathbb{R}^m$ is transverse to $\{0\}$. Therefore, we have the residual set

$$\mathcal{F}_4 = \{g_S \in C^k(S, \mathbb{R}^m) : \Gamma_R(\cdot \mid g_S) \pitchfork \{0\}\}, \tag{56}$$

and when $m > 2n, \forall g_S \in \mathcal{F}_4$ makes $f$ injective on $\Omega_R$.

### *Injectivity on mixed $SR$-side:*

Finally, consider the mixed part modulo the equivalent point defined by $x \sim r(x)$:

$$\Omega_{SR} := (U_S \times U_R) \setminus \Big\{((x, 0), (r(x), 0)) : x \in S\Big\}. \tag{57}$$

For $x \in S, y \in r(S)$ and $t, s \in [0, \epsilon)$, define the total map

$$\Gamma_{SR}(((x, t), (y, s)), g_S) := (f_S(x) + t\,g_S(x)) - (f_R(y) + s\,g_R(y)). \tag{58}$$

Using $f_R(r(x)) = f_S(x)$ and (31), we introduce $\tilde{x} := r^{-1}(x)$ and write for $y = r(\tilde{x})$:

$$\Gamma_{SR}((x, t), (r(\tilde{x}), s), g_S) = (f_S(x) + t\,g_S(x)) - (f_S(\tilde{x}) + s(\alpha(\tilde{x})g_S(\tilde{x}) + c(\tilde{x}))) \tag{59}$$

with $\alpha(\tilde{x}) = \frac{\nu_S(\tilde{x})}{\nu_R(r(\tilde{x}))} < 0$ by (30) and Assumption 2.

When $(t, s) = (0, 0)$ and $x \neq \tilde{x}$, as $f_S(\cdot) = g(\cdot)$ is an embedding, $\Gamma_{SR}(((x, 0), (r(\tilde{x}), 0)), g_S) \neq 0$.

When we apply differential on the parametric $g_S$ to the total map, for $h \in C^k(S, \mathbb{R}^m)$ we have:

$$D_{g_S}\Gamma_{SR}[h] = th(x) - s\alpha(\tilde{x})h(\tilde{x}), h \in C^k(S, \mathbb{R}^m). \tag{60}$$

When $(t, s) \neq (0, 0)$ and $x \neq \tilde{x}$, (60) can achieve any value by varying $h$ thus is immersion that ensures $\Gamma_{SR} \pitchfork \{0\}$.

When $x = \tilde{x}$, we note that Assumption 2 implies $t - s\alpha(x) > 0$ by the $\alpha(x) < 0$ and $t, s > 0$. Thus we have $h \mapsto (t - s\alpha(x))h(x)$ can achieve any value in $\mathbb{R}^m$, which conforms that (60) is surjectiv everywhere and thus $\Gamma_{SR} \pitchfork \{0\}$.

Finally, we have the residual set

$$\mathcal{F}_5 = \{g_S \in C^k(S, \mathbb{R}^m) : \Gamma_{SR}(\cdot \mid g_S) \pitchfork \{0\}\}, \tag{61}$$

such that $\forall g_S \in \mathcal{F}_5$ makes $f$ injective on $\Omega_R$. Thus, by Theorem 3, we have the dimension of the manifold $\Gamma_{SR}^{-1}(\cdot \mid g_S)$:

$$\dim \Gamma_{SR}^{-1}(\cdot \mid g_S) = \dim \Omega_{SR} - m = 2n - m. \tag{62}$$

When $m > 2n$, $\Gamma_{SR}^{-1}(\cdot \mid g_S) = \emptyset$ for $\forall g_S \in \mathcal{F}_5$, which ensures $f$ is injective on $\Omega_{SR}$.

Finally, we have the residual set

$$\mathcal{F}_{\text{inj}} := \mathcal{F}_3 \cap \mathcal{F}_4 \cap \mathcal{F}_5$$

where $\forall g_S \in \mathcal{F}_{\text{inj}}$ the $f$ is injective on the collar neighborhood $U$. $\qquad\square$

### B.2.3. SUMMARY

Finally, we conclude the result of the local property of $f$ on $U$:

**Remark 7.** *By Lemma 3 and Lemma 4, we have that $\mathcal{F}_{\text{imm}} \cap \mathcal{F}_{\text{inj}}$ is a residual set, and thus a generic choice of $g_S$ guarantees that $f$ defined in (34) is both an immersion and is injective on $U$ whenever $m > 2n$.*

### B.3. Global Extension by Relative Perturbation

In the last subsection, we have proved that whenever $m > 2n$, the function $f$ is generically an immersion and an injective function on $U := U_S \cup U_R$. Now we show that this property can be extended to the entire state space.

Let $K := S \cup R$. Choose an open set $U_0 \Subset U$ such that $K \Subset U_0 \Subset U$. Choose a smooth cutoff $\rho : M \to [0, 1]$ with $\rho \equiv 1$ on $\overline{U}_0$ and $\text{supp}(\rho) \subset U$, and fix any $C^k$ map $\phi : M \to \mathbb{R}^m$. Define $f_0 : M \to \mathbb{R}^m$:

$$f_0(x) := \begin{cases} \rho(x)\, f(x) + (1 - \rho(x))\, \phi(x), & x \in U, \\ \phi(x), & x \notin U. \end{cases} \tag{63}$$

Then $f_0 \in C^k(M, \mathbb{R}^m)$ and $f_0 \equiv f$ on $\overline{U}_0$.

We choose the buffer neighborhood $U_1$ as a smooth domain with boundary. Specifically, since $K \Subset U_0$, choose a smooth cutoff function $\chi : M \to [0, 1]$ such that $\chi \equiv 1$ on a neighborhood of $K$, $\text{supp}(\chi) \Subset U_0$. By Sard's theorem, choose a regular value $c \in (0, 1)$ of $\chi$, and define $U_1 := \{x \in M : \chi(x) > c\}$. Then $K \subset U_1 \Subset U_0 \Subset U$, and $\partial U_1 = \chi^{-1}(c)$ is a smooth embedded hypersurface. Equivalently, $U_1 = \{\chi > c\}$, $\overline{U}_1 = \{\chi \geq c\}$, $\partial U_1 = \{\chi = c\}$. Thus $U_1$ is a smooth domain with boundary.

We restrict perturbations of $f_0$ to vanish on $U_1$. Since the perturbations are continuous, this also implies that they vanish on $\overline{U}_1$. Thus all transversality arguments are carried out away from $\partial U_0$; the only additional boundary stratum to consider is $\partial U_1$.

Define the linear subspace of perturbations vanishing on $U_1$,

$$\mathcal{P} := \{h \in C^k(M, \mathbb{R}^m) : h|_{U_1} = 0\}, \tag{64}$$

and consider the family

$$f_h := f_0 + h, \qquad h \in \mathcal{P}. \tag{65}$$

Then $f_h \equiv f_0$ on $\overline{U}_1$ because $h|_{U_1} = 0$ and $h$ is continuous. Hence the immersion on $\overline{U}_1$ and injectivity on $\overline{U}_1/\sim$ are preserved *exactly* for $f_h$. Then we proceed to show that a generic choice of $h$ ensures the property of $f$ on $\overline{U}_1$ can be extended to the entire $M$.

### B.3.1. RELATIVE GLOBAL IMMERSION

**Lemma 5** (Relative global immersion). *Assume $m > 2n - 1$. There exists a residual subset $\mathcal{P}_{\text{imm}} \subset \mathcal{P}$ such that for all $h \in \mathcal{P}_{\text{imm}}$, the map $f_h = f_0 + h$ is an immersion on $M$ and satisfies $f_h \equiv f$ on $\overline{U}_1 \Subset U$.*

*Proof.* Let $J^1(M, \mathbb{R}^m)$ be the 1-jet bundle and write

$$j^1 f_h(x) = (x, f_h(x), Df_h(x)). \tag{66}$$

For $k \leq n - 1$, let $\Sigma_k \subset J^1(M, \mathbb{R}^m)$ denote the rank-$k$ stratum of 1-jets, i.e., those with $\mathrm{rank}(Df_h(x)) = k$:

$$\Sigma_k := \left\{ (x, y, A) \in J^1(M, \mathbb{R}^m) : \mathrm{rank}(A) = k \right\}. \tag{67}$$

Since the projection $\pi_A : J^1(M, \mathbb{R}^m) \to \mathbb{R}^{m \times n}$, $\pi_A(x, y, A) = A$, is a smooth submersion with $d\pi_A(dx, dy, dA) = dA$ surjective at every point, and the rank-$k$ matrix set $R_k := \{A \in \mathbb{R}^{m \times n} : \mathrm{rank}(A) = k\}$ is a smooth embedded submanifold of $\mathbb{R}^{m \times n}$, it follows by the preimage theorem that $\Sigma_k = \pi_A^{-1}(R_k)$ is a smooth embedded submanifold of $J^1(M, \mathbb{R}^m)$. Moreover, $\dim R_k = k(m + n - k)$, hence $\mathrm{codim}\, \Sigma_k = \mathrm{codim}\, R_k = mn - k(m + n - k)$.

The immersion-failure set is the determinantal set $\Sigma_{<n} := \bigcup_{k \leq n-1} \Sigma_k$, where the smallest codimension occurs on the rank-$(n-1)$ stratum $\Sigma_{n-1}$, which is $\mathrm{codim}\, \Sigma_{n-1} = m - n + 1$.

Define the total 1-jet map:

$$\begin{aligned} J : (M \setminus \overline{U}_1) \times \mathcal{P} &\to J^1(M, \mathbb{R}^m) \\ J(x, h) &:= j^1 f_h(x) \end{aligned}. \tag{68}$$

We claim that $J \pitchfork \Sigma_{n-1}$ on $(M \setminus \overline{U}_1) \times \mathcal{P}$.

Indeed, fix $(x, h)$ with $x \in M \setminus \overline{U}_1$ and choose a coordinate chart $\varphi : B \to \mathbb{R}^n$ with $p \in B \subset M \setminus \overline{U}_1$ and $\varphi(x) = 0$. For any $(a, A) \in \mathbb{R}^m \times \mathbb{R}^{m \times n}$, pick $\rho \in C_c^\infty(B)$ with $\rho(x) = 1$ and $D\rho(x) = 0$, and define $h' \in \mathcal{P}$ by

$$h'(q) := \rho(q)\,(a + A\,\varphi(q)), \qquad q \in M, \tag{69}$$

extended by 0 outside $B$. Then $h'|_{U_1} = 0$ and we can verify that $h'$ reaches value

$$h'(x) = a, \qquad Dh'(x) = A \circ D\varphi(x). \tag{70}$$

Since $J(x, h) = j^1 f_h(x) = (x, f_h(x), Df_h(x))$, in order to verify transversality it is sufficient to consider variations in the parameter $h$ while keeping $x$ fixed. For the curve $t \mapsto (x, h + th')$, we have

$$\left.\frac{d}{dt}\right|_{t=0} J(x, h + th') = (0,\ h'(x),\ Dh'(x)) \in T_{(x, f_h(x), Df_h(x))} J^1(M, \mathbb{R}^m). \tag{71}$$

By the above construction, the pair $(h'(x), Dh'(x))$ can be chosen arbitrarily in $\mathbb{R}^m \times \mathbb{R}^{m \times n}$. Since the rank constraint defining $\Sigma_{n-1}$ involves only the third component, and the $(x, y)$-directions are unconstrained, these parameter variations already suffice to establish $J \pitchfork \Sigma_{n-1}$.

Therefore $D_h J_{(x,h)}$ is surjective onto the $(f, Df)$-components at $x$. Since the rank constraint defining $\Sigma_{n-1}$ imposes no restriction on the $(x, y)$-components, we have the first two components surjective by the derivative of $x$. By the fact that $J^1(M, \mathbb{R}^m) \cong M \times \mathbb{R}^m \times \mathbb{R}^{m \times n}$, $\Sigma_{n-1} = M \times \mathbb{R}^m \times R_{n-1}$, we have:

$$T_{(x, y, A)} \Sigma_{n-1} = T_x M \oplus \mathbb{R}^m \oplus T_A R_{n-1}, \tag{72}$$

Thus, for all $(x, h) \in J^{-1}(\Sigma_{n-1})$,

$$\mathrm{Im}(dJ_{(x,h)}) + T_{J(x,h)} \Sigma_{n-1} = T_{J(x,h)} J^1(M, \mathbb{R}^m), \tag{73}$$

so $J \pitchfork \Sigma_{n-1}$ by Proposition 1.

By Theorem 4, for each $k \leq n - 1$ the set

$$\mathcal{P}_k := \{h \in \mathcal{P} : j^1 f_h \pitchfork \Sigma_k \text{ on } M \setminus \overline{U}_1\} \tag{74}$$

is residual in $\mathcal{P}$. For such $h$ the preimage $(j^1 f_h)^{-1}(\Sigma_k)$ is an embedded submanifold of $M \setminus \overline{U}_1$ and, by Theorem 3,

$$\dim(M \setminus \overline{U}_1) - \mathrm{codim}\, \Sigma_{n-1} = n - (m - n + 1) = 2n - m - 1. \tag{75}$$

If $m > 2n - 1$ (in particular if $m > 2n$), this dimension is negative, hence $(j^1 f_h)^{-1}(\Sigma_{n-1}) = \emptyset$. Since each lower-rank stratum $\Sigma_k$ ($k \leq n - 2$) has strictly larger codimension than $\Sigma_{n-1}$, the same dimension count also rules out $(j^1 f_h)^{-1}(\Sigma_k)$ when $m > 2n - 1$.

Finally, we conclude that there exists the residual set

$$\mathcal{P}_{\text{imm}} := \bigcap_{k=0}^{n-1} \mathcal{P}_k \tag{76}$$

such that for all $f_h \in \mathcal{P}_{\text{imm}}$ the map $f_h$ is an immersion on $M \setminus \overline{U}_1$. Together with $f_h \equiv f$ on $U_1$ and the immersion property already established on $U_1 \subset U$, we conclude that $f_h$ is an immersion on all of $M$. $\square$

### B.3.2. RELATIVE GLOBAL INJECTIVITY

**Lemma 6** (Relative global injectivity). *Assume $m > 2n$. There exists a residual subset $\mathcal{P}_{\text{inj}} \subset \mathcal{P}$ such that for all $h \in \mathcal{P}_{\text{inj}}$, the perturbed map $f_h = f_0 + h$ introduces no new identifications, i.e.,*

$$f_h(x) = f_h(y) \implies x \sim y, \quad \forall x, y \in M/\sim .$$

*Proof.* As $f_h$ is already fixed on $\overline{U}_1$, we identify the violation of injectivity for the case with at least one point outside on $M \setminus \overline{U}_1$. Let $\Delta = \{(x, y) | x = y, x, y \in M\} \subset M \times M$, define the open sets for two or one points outside $\overline{U}_1$ as:

$$\begin{aligned} \Omega_{\text{oo}} &:= \left((M \setminus \overline{U}_1) \times (M \setminus \overline{U}_1)\right) \setminus \Delta, \\ \Omega_{\text{io}} &:= U_1 \times (M \setminus \overline{U}_1) \end{aligned} \tag{77}$$

and the case with one point on the boundary of

$$\Omega_{\partial\text{o}} := \partial U_1 \times (M \setminus \overline{U}_1) \tag{78}$$

Define the total difference maps for $* = oo, io$ as

$$\Gamma_{(\cdot)} : \Omega_{(\cdot)} \times \mathcal{P} \to \mathbb{R}^m, \Gamma_{(\cdot)}((x, y), h) := f_h(x) - f_h(y), \tag{79}$$

The we show that $\Gamma_{(\cdot)} \pitchfork \{0\}$ by Proposition 1.

For $\Gamma_{\text{oo}}$, when $x \neq y$ and $x, y \in M \setminus \overline{U}_1$, the differential of the parametric direction is

$$D_h \Gamma_{\text{oo}}[h'] = h'(x) - h'(y). \tag{80}$$

By choosing $h' \in \mathcal{P}$ supported in two small disjoint coordinate balls centered at $x$ and $y$ (both contained in $M \setminus \overline{U}_1$), and using standard bump functions in each chart, the values $h'(x)$ and $h'(y)$ can be prescribed independently and arbitrarily in $\mathbb{R}^m$. Hence (80) is surjective onto $\mathbb{R}^m$. Thus, there exist a residual set

$$\mathcal{P}_{\text{oo}} = \{h \in \mathcal{P} : \Gamma_{\text{oo}}(\cdot \mid h) \pitchfork \{0\}\} \tag{81}$$

For $\Gamma_{\text{io}}$, at any zero with $(x, y) \in U_1 \times (M \setminus \overline{U}_1)$ we have $h'(x) = 0$ for all $h' \in \mathcal{P}$, hence

$$D_h \Gamma_{\text{io}}[h'] = -h'(y), \tag{82}$$

which is surjective by choosing $h'$ supported near $y \in M \setminus \overline{U}_1$. Thus, we have the residual set

$$\mathcal{P}_{\text{io}} = \{h \in \mathcal{P} : \Gamma_{\text{io}}(\cdot \mid h) \pitchfork \{0\}\}. \tag{83}$$

For $\Gamma_{\partial\text{o}}$, we need to verify the transversality condition on $(x, y) \in \partial U_1 \times (M \setminus \overline{U}_1)$. Similar to the $\Gamma_{\text{io}}$ case, we have $h'(x) = 0$ on $x \in \partial \overline{U}_1$. Thus, similar to $D_h \Gamma_{\text{io}}$, we have $D_h \Gamma_{\partial\text{o}}[h'] = -h'(y)$, which is surjective guaranteed by the $h \in \mathcal{P}$. We note that the surjectivity in the parameter is sufficient for transversality that ensures the total map for the version of Theorem 4 with boundary is also transversal. Thus, we have the residual set

$$\mathcal{P}_{\partial\text{o}} = \{h \in \mathcal{P} : \Gamma_{\text{io}}(\cdot | h) \pitchfork \{0\}\}. \tag{84}$$

Applying Theorem 3, there exsits a residual set:

$$\mathcal{P}_{\text{inj}} := \mathcal{P}_{\text{oo}} \cap \mathcal{P}_{\text{io}} \cap \mathcal{P}_{\partial\text{o}}, \tag{85}$$

such that $\forall h \in \mathcal{P}_{\text{inj}}$, each slice $\Gamma_{(\cdot)}(\cdot, h) \pitchfork \{0\}$, and $\Gamma_{(\cdot)}^{-1}(0 \mid h)$ set is a submanifold of dimension:

$$\dim \Gamma_{(\cdot)}^{-1}(0 \mid h) = \dim \Omega_{(\cdot)} - m = 2n - m. \tag{86}$$

If $m > 2n$, all these dimensions are negative, hence $\Gamma_{(\cdot)}^{-1}(0 \mid h) = \emptyset$. Therefore, such a generic $h$ will not introduce new collisions. $\qquad\square$

### B.3.3. SUMMARY

Finally, we conclude the results of the subsection by

**Remark 8.** *By Lemma 5 and 6, we conclude that there is a residual set:*

$$\mathcal{P}' := \mathcal{P}_{\text{imm}} \cap \mathcal{P}_{\text{inj}} \subset \mathcal{P}, \tag{87}$$

*such that $\forall h \in \mathcal{P}'$, $f_h := f_0 + h$ is an immersion and injective on $M/\sim$.*

### B.4. Induced Embedding

We note show that the $f : M \to \mathbb{R}^m$ constructed by the first-order collar extension and the relative perturbation induces an embedding on $M_{\mathcal{H}}$.

**Lemma 7** (Induced embedding on (Hybrifold))**.** *Let $f : M \to \mathbb{R}^m$ satisfy (C-1) and (C-2), and suppose that $f$ is injective and an immersion on $M/\sim$. Let $\pi : M \to M_{\mathcal{H}} = M/\sim$ be the quotient map induced by the relation $x \sim r(x)$ for all $x \in S$. Then there exists a unique continuous map*

$$\bar{f} : M_{\mathcal{H}} \to \mathbb{R}^m$$

*such that $\bar{f} \circ \pi = f$. Moreover, $\bar{f}$ is injective, and if $M_{\mathcal{H}}$ is compact, then $\bar{f}$ is a topological embedding.*

*Proof.* By (C-1), $f$ is constant on equivalence classes of $\sim$. Hence the map

$$\bar{f}([x]) := f(x), \qquad [x] \in M_{\mathcal{H}},$$

is well-defined and satisfies $\bar{f} \circ \pi = f$. Uniqueness follows from the surjectivity of $\pi$. Since $\pi$ is a quotient map and $f$ is continuous, $\bar{f}$ is continuous.

If $\bar{f}([x]) = \bar{f}([y])$, then $f(x) = f(y)$, which implies $[x] = [y]$ by injectivity of $f$ on $M$. Thus $\bar{f}$ is injective. If $M_{\mathcal{H}}$ is compact, then a continuous injective map into the Hausdorff space $\mathbb{R}^m$ is a topological embedding, completing the proof. $\qquad\square$

### B.5. Summary

Finally, we conclude that there exist a generic construction of $f \in C^k(M, \mathbb{R}^m)$ that satisfies (C-1), (C-2), and (C-3) whenever $m > 2n$:

**Remark 9** (Genericity of the construction)**.** *Assume $m > 2n$. The map $f$ in Theorem 6 is obtained by first constructing $f$ on a collar neighborhood from parameters $(g, g_S)$ and then extending it to a global map $f_0$ which is perturbed as $f_h := f_0 + h$. The desired properties (C-1), (C-2), and (C-3) hold for a* generic *choice of the construction parameters, in the following sense:*

1. (Generic choice of $g$) *Since $\dim S = n - 1$ and $m > 2n$ (hence $m > 2 \dim S$), one may choose $g \in C^k(S, \mathbb{R}^m)$ generically (i.e., from a residual subset) so that $g : S \to \mathbb{R}^m$ is an embedding.*
2. (Generic collar embedding) *Choose $g_S$ from the residual set $\mathcal{F}_{\text{imm}} \cap \mathcal{F}_{\text{inj}} \subset C^k(S, \mathbb{R}^m)$ (with $g : S \to \mathbb{R}^m$ chosen as an embedding). Then the collar construction yields a map $f$ satisfying (C-1)–(C-2) and is an immersion and injective on a neighborhood of $S$.*

3. *(Generic relative perturbation)* *With $U_0 \Subset U$ fixed and $\mathcal{P} := \{h \in C^k(M, \mathbb{R}^m) : h|_{U_1} = 0\}$, choose $h$ from the residual set $\mathcal{P}' \subset \mathcal{P}$. Then $f_h = f_0 + h$ is a global immersion and introduces no new identifications, so that the induced map on the quotient is an embedding, i.e., (C-3) holds.*

*In particular, selecting $(g, g_S, h)$ from these residual sets (thus a generic construction) produces a map $f$ satisfying (C-1), (C-2), and (C-3).*

## C. Comparison with Existing Methods

In this appendix, we summarize the experiment setups for all the baselines and the proposed method.

### C.1. Experiment Setups

The experiment setup for the Torus, Klein Bottle, and Bouncing Ball examples.

Table 5: Model parameters.

|  | Proposed | (Teng et al., 2025) | (Chen et al., 2018) | (Lusch et al., 2018) |
|---|---|---|---|---|
| Vector Field | $[64]\times2$ | $[64]\times2$ | $[128]\times2$ | Linear |
| Encoder | $[64]\times3$ | $[64]\times3$ | N/A | [64,128,256] |
| Decoder | $[128]\times3$ | $[128]\times8$ | N/A | [256,128,64] |
| Vector Field Dim. | $2n$ | $2n$ | $n$ | 256 |

The experiment setup for the Three-Link Walker Examples

Table 6: Model parameters.

|  | Proposed | (Teng et al., 2025) | (Chen et al., 2018) | (Lusch et al., 2018) |
|---|---|---|---|---|
| Vector Field | $[64]\times3$ | $[64]\times3$ | $[128]\times2$ | Linear |
| Encoder | $[64]\times3$ | $[64]\times3$ | N/A | [64,128,256] |
| Decoder | $[128]\times3$ | $[128]\times8$ | N/A | [256,128,64] |
| Vector Field Dim. | $2n$ | $2n$ | $n$ | 256 |

The experiment setup for the 3D Bouncing Ball Examples

Table 7: Model parameters.

|  | Proposed | (Teng et al., 2025) | (Chen et al., 2018) | (Lusch et al., 2018) |
|---|---|---|---|---|
| Vector Field | $[256]\times2$ | $[64]\times2$ | $[128]\times2$ | Linear |
| Encoder | $[256]\times3$ | $[64]\times3$ | N/A | [64,128,256] |
| Decoder | $[256]\times3$ | $[128]\times8$ | N/A | [256,128,64] |
| Vector Field Dim. | $2n$ | $2n$ | $n$ | 256 |

The setup for the Event Neural ODE (Chen et al., 2018) and Neural ODE with RNN encoder (Rubanova et al., 2019)

Table 8: Hidden layers for each component of the event Neural ODE (Chen et al., 2018).

|  | Vector Field | Guard | Reset Function | Vector Field Dim. |
|---|---|---|---|---|
| Event ODE | $[256]\times2$ | $[256]\times3$ | $[256]\times3$ | $n$ |

Table 9: Hidden layers for each component of the Neural ODE (RNN) (Rubanova et al., 2019).

|  | Vector Field | Encoder | Decoder | Vector Field Dim. |
|---|---|---|---|---|
| ODE (RNN) | $[100]\times2$ | GRU(100) | Linear$\times3$ | 20 |

The we consider a $w_z = w_x = 10$ and $w_c = 1000$ as the weights for the proposed method. The threshold for the latent covariance is $\Lambda = 0.09$.

The batch size for all the experiments is 2048. The curriculum for our method is rolling out [32, 64, 128, 200] time steps ahead for [2000, 2000, 2000, 4000] steps of gradient descent. For 3D Bouncing Ball, we do [3000, 3000, 3000, 6000] steps of gradient descent.

### C.2. Visualizations

We provide the visualizations for the proposed methods and the baselines. We plot the trajectories of each DOF with respect to time and the latent trajectories for (Teng et al., 2025) as it is closely related to our method with a differential

topology-based latent design. We note that Chyll indicates (Teng et al., 2025), Neural ODE indicates (Chen et al., 2018), Latent ODE (RNN) indicates (Rubanova et al., 2019) and Koopman indicates (Lusch et al., 2018).

## C.2.1. BOUNCING BALL

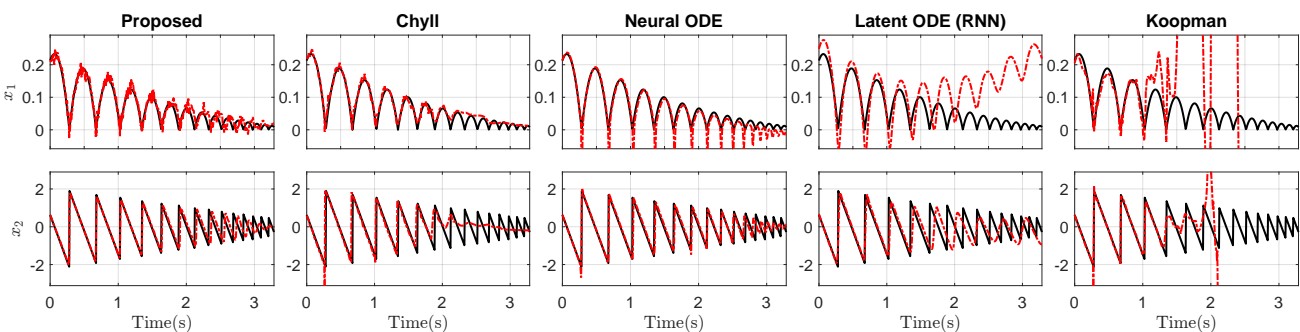

Figure 6: Prediction of $x$ trajectories of Bouncing Ball.

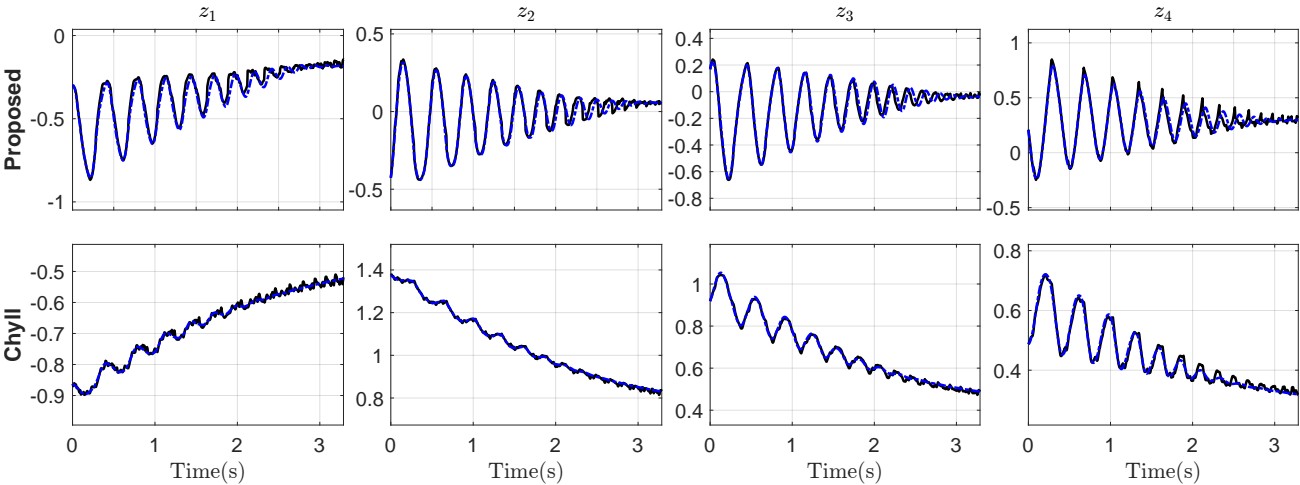

Figure 7: Prediction of latent trajectories of Bouncing Ball.

## C.2.2. TORUS

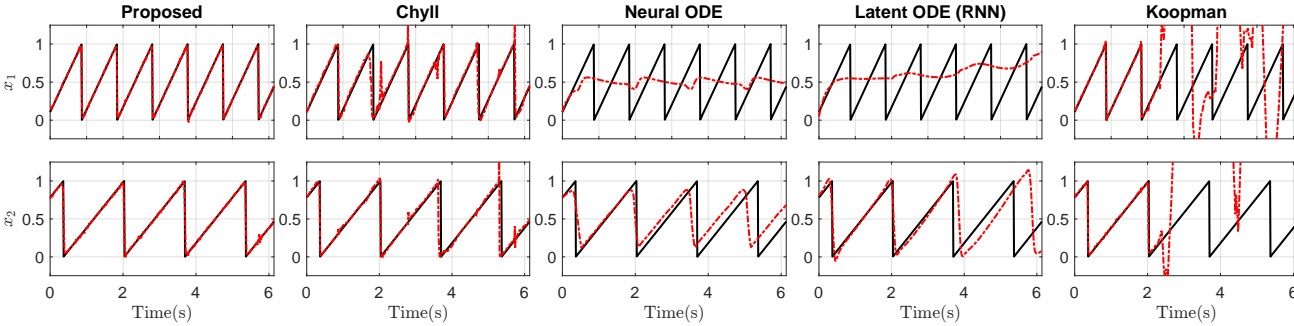

Figure 8: Prediction of $x$ trajectories of Torus.

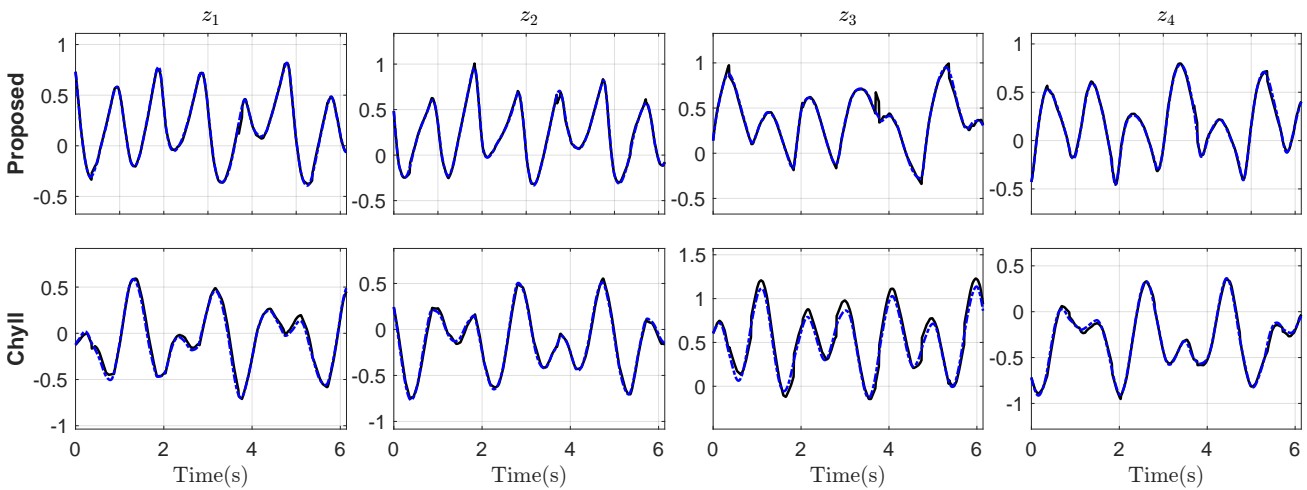

Figure 9: Prediction of latent trajectories of Torus.

### C.2.3. KLEIN BOTTLE

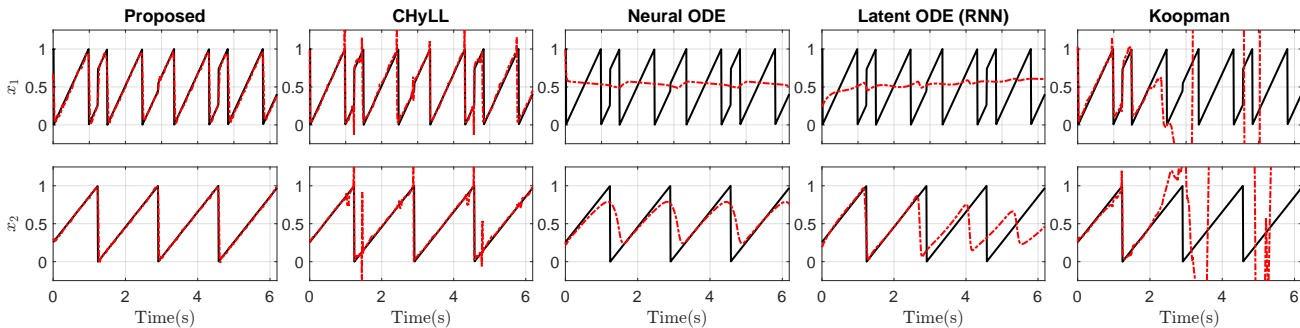

Figure 10: Prediction of $x$ trajectories of Klein Bottle.

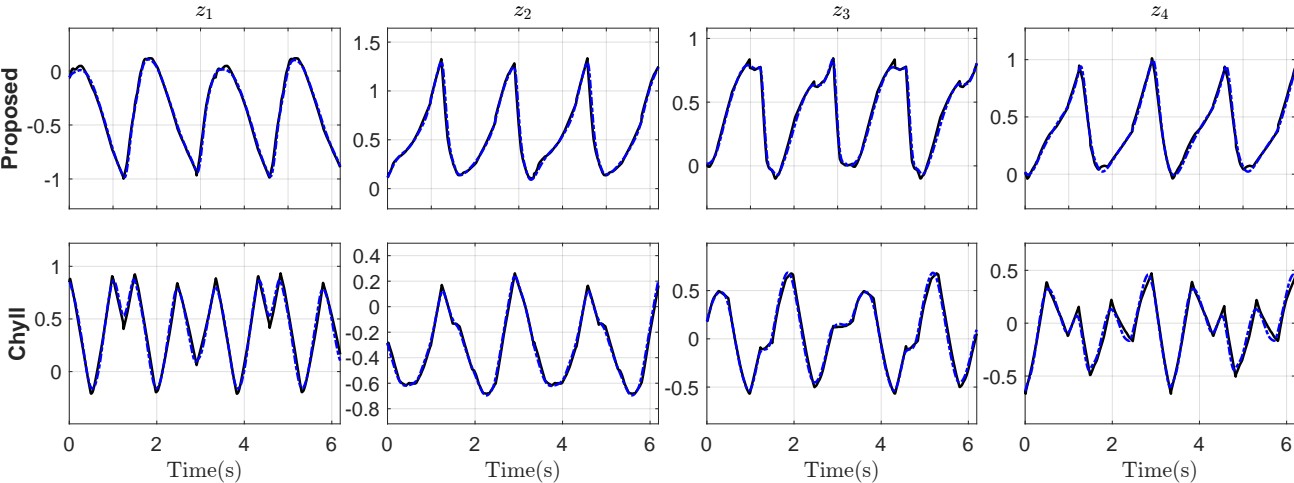

Figure 11: Prediction of latent trajectories of Klein Bottle.

C.2.4. THREE-LINK WALKER

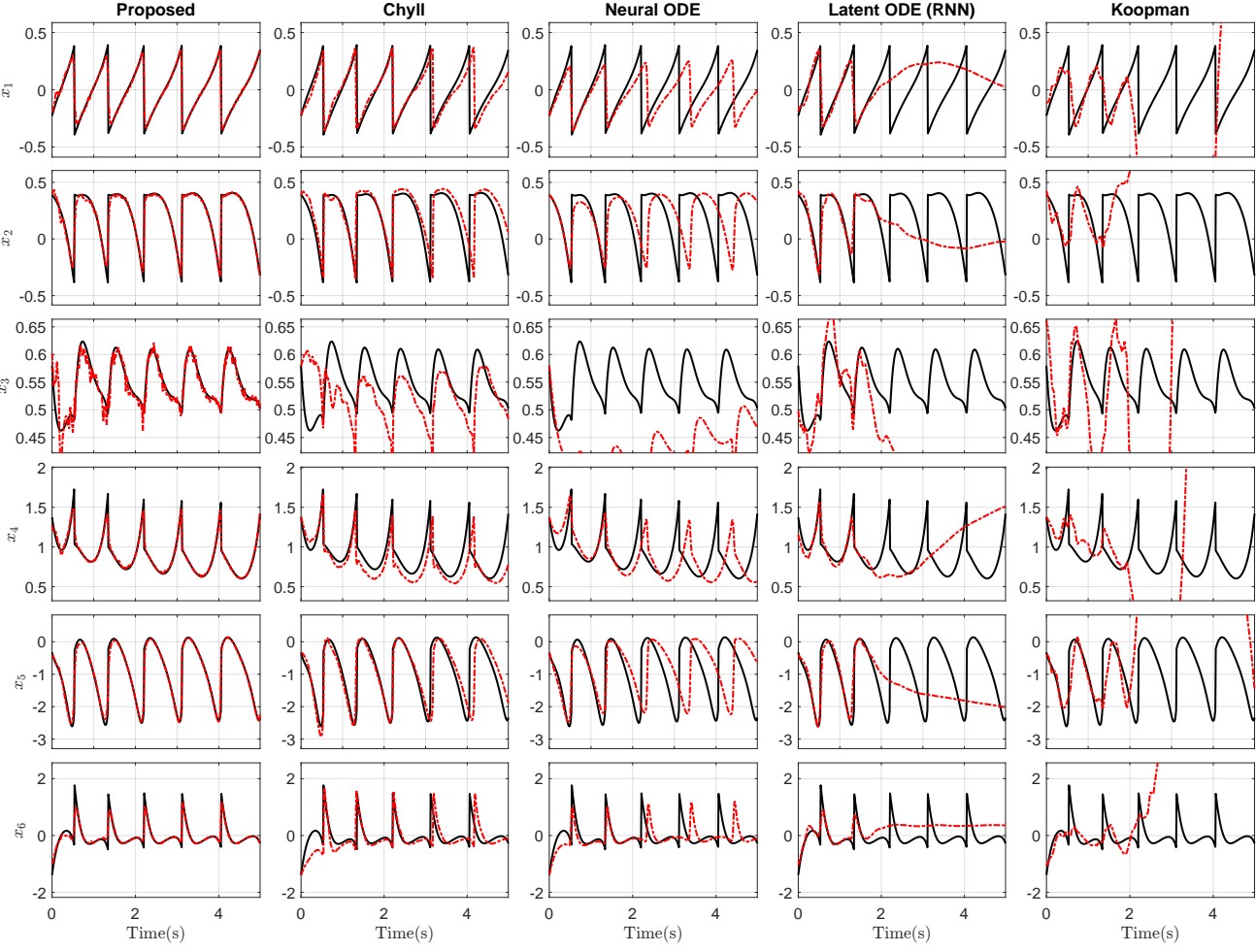

Figure 12: Prediction of $x$ trajectories of Three-Link Walker.

# Proposed

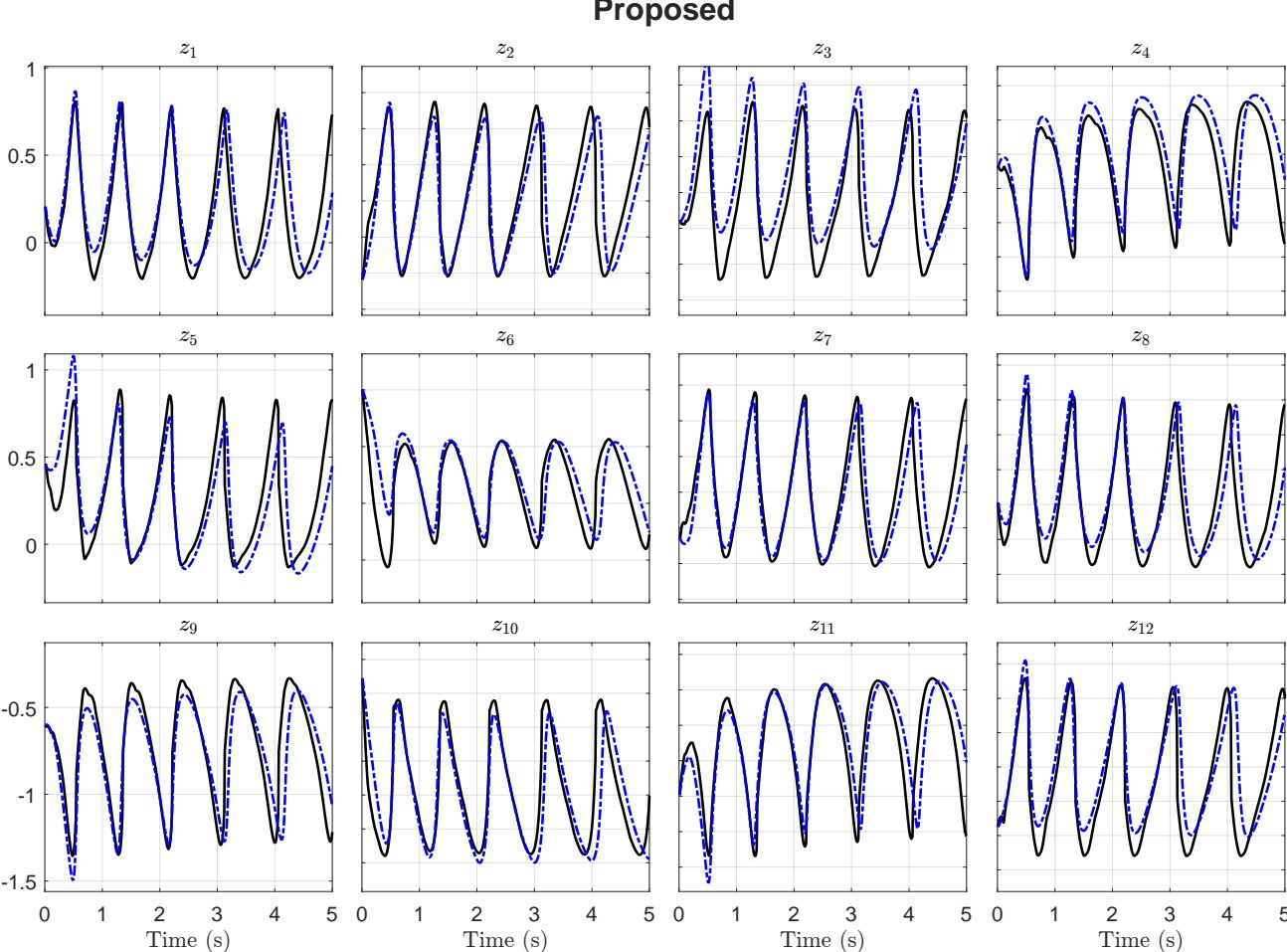

Figure 13: Prediction of latent trajectories of Klein Bottle for the proposed method.

## Chyll

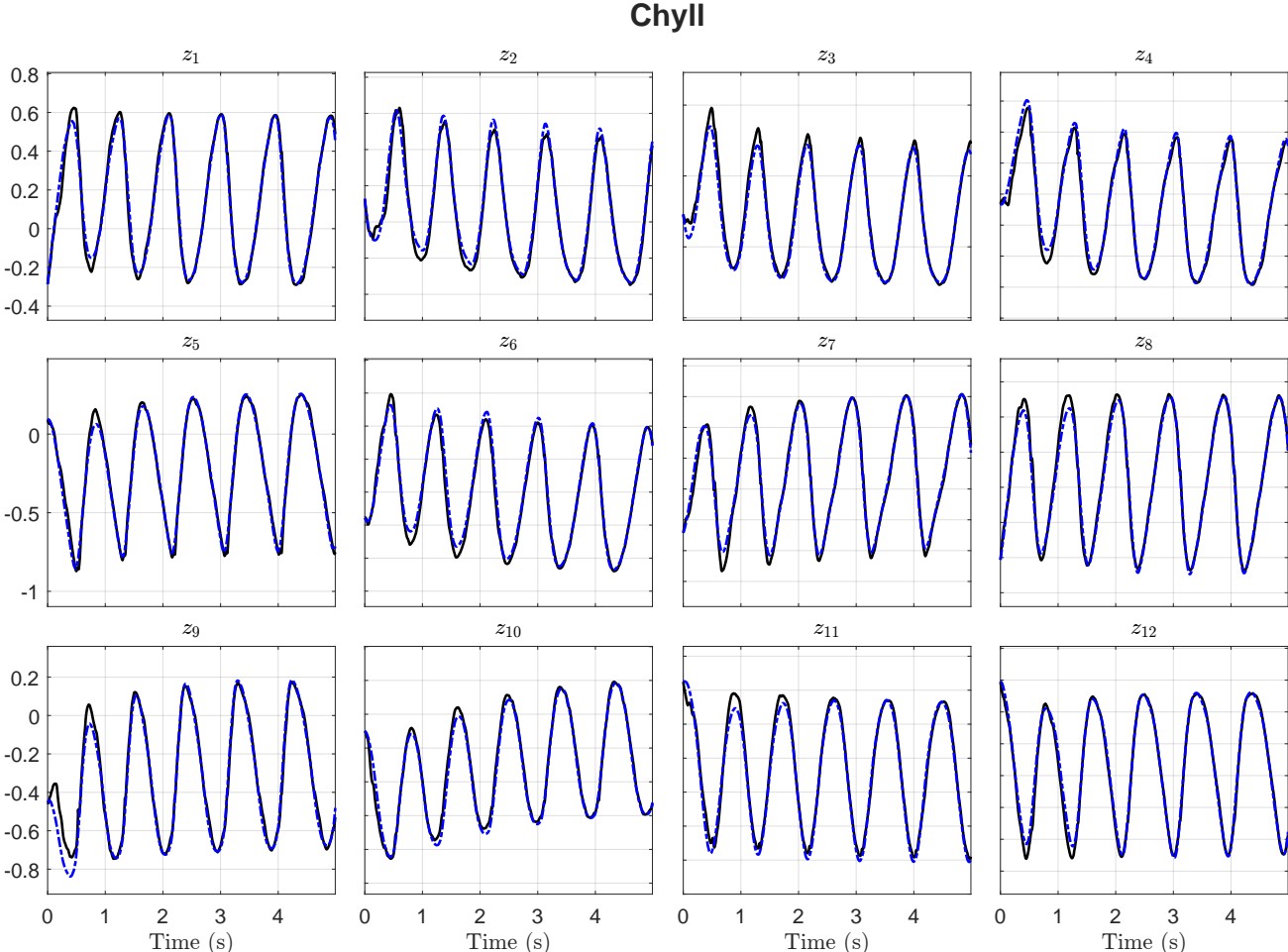

Figure 14: Prediction of latent trajectories of Klein Bottle for (Teng et al., 2025).

C.2.5. 3D Bouncing Ball

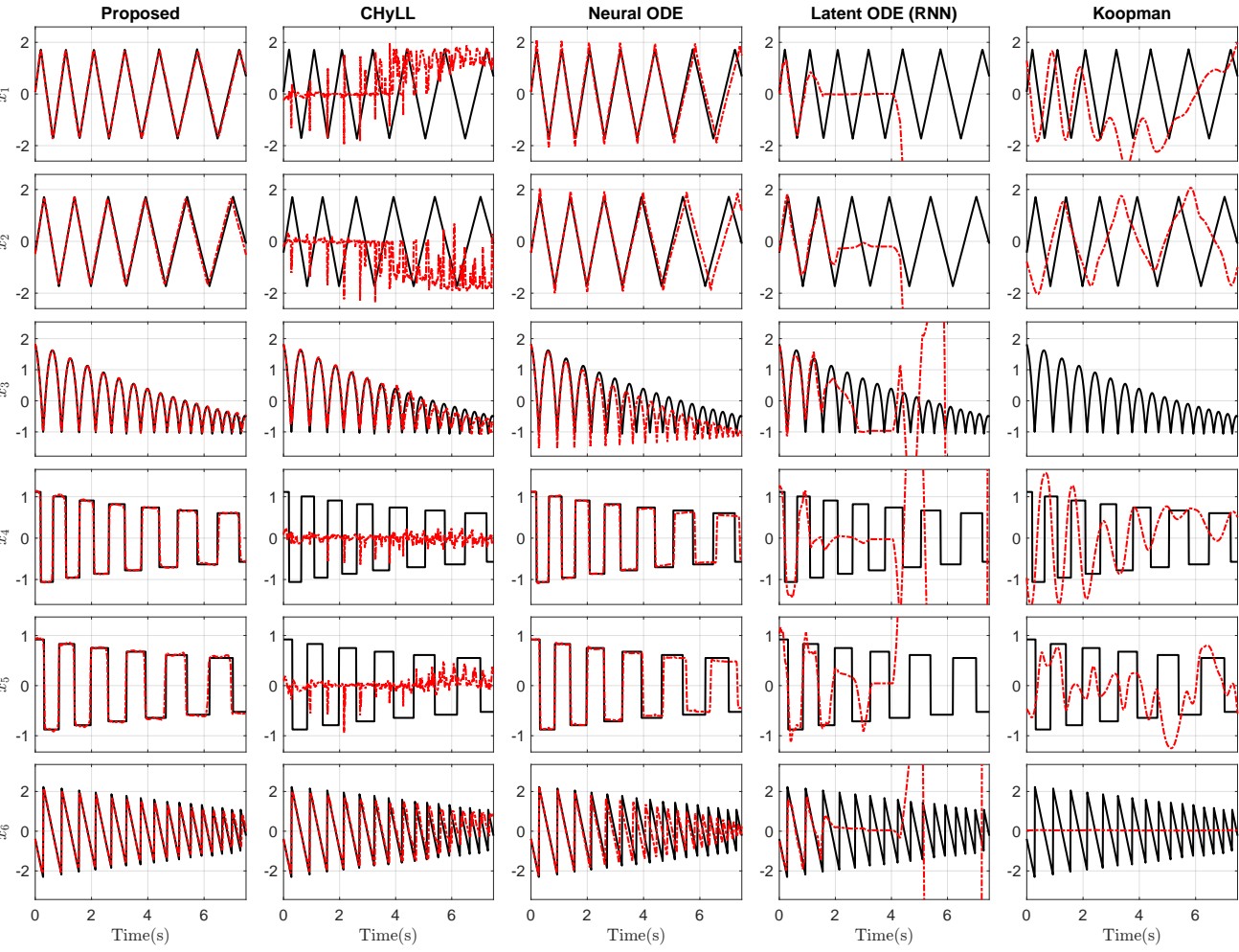

Figure 15: Prediction of $x$ trajectories of 3D Bouncing Ball. From the top to bottom: position $x - y - z$ and the velocity $x - y - z$.

## Proposed

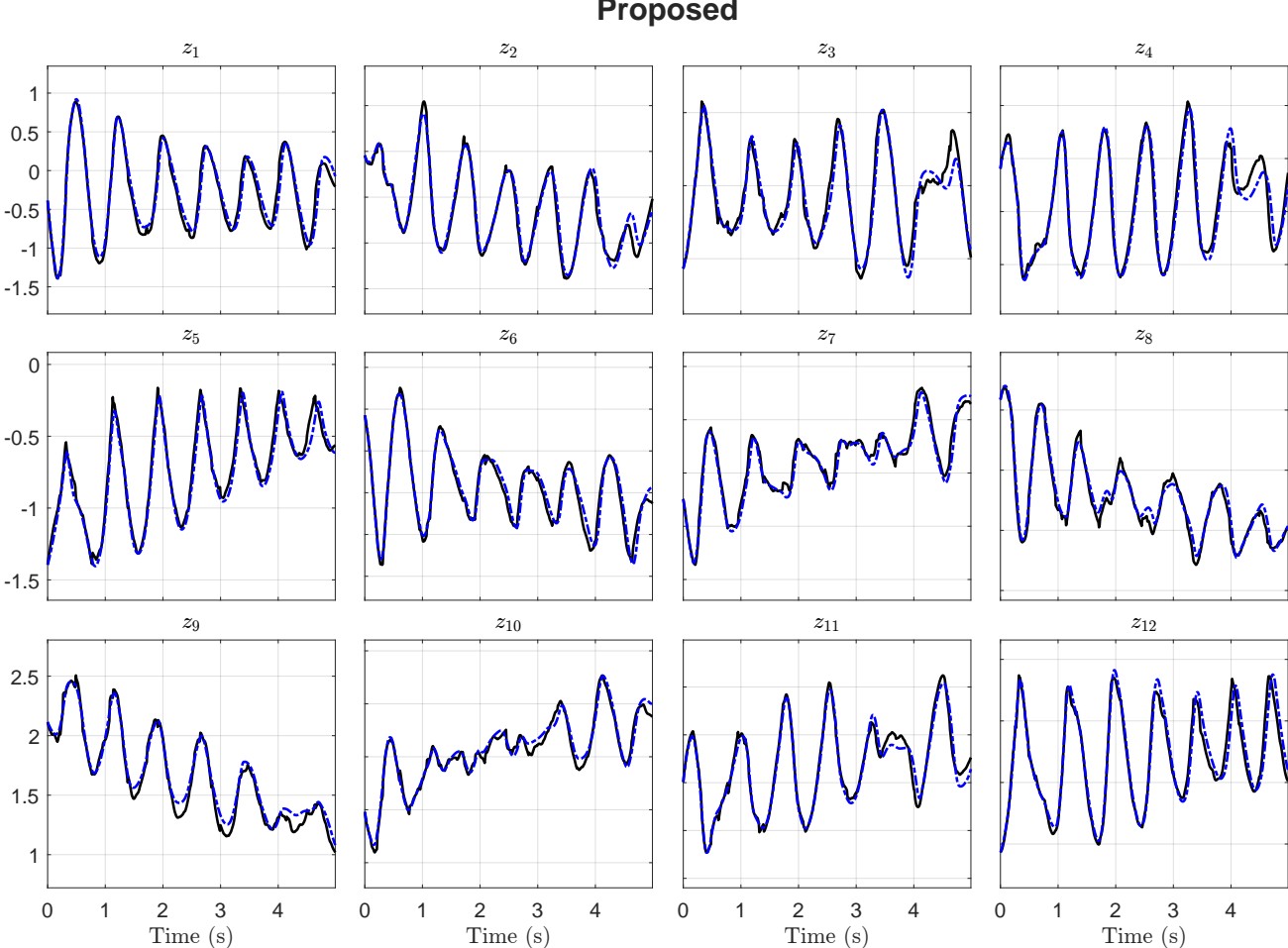

Figure 16: Prediction of latent trajectories of 3D Bouncing Ball for the proposed method.

## Chyll

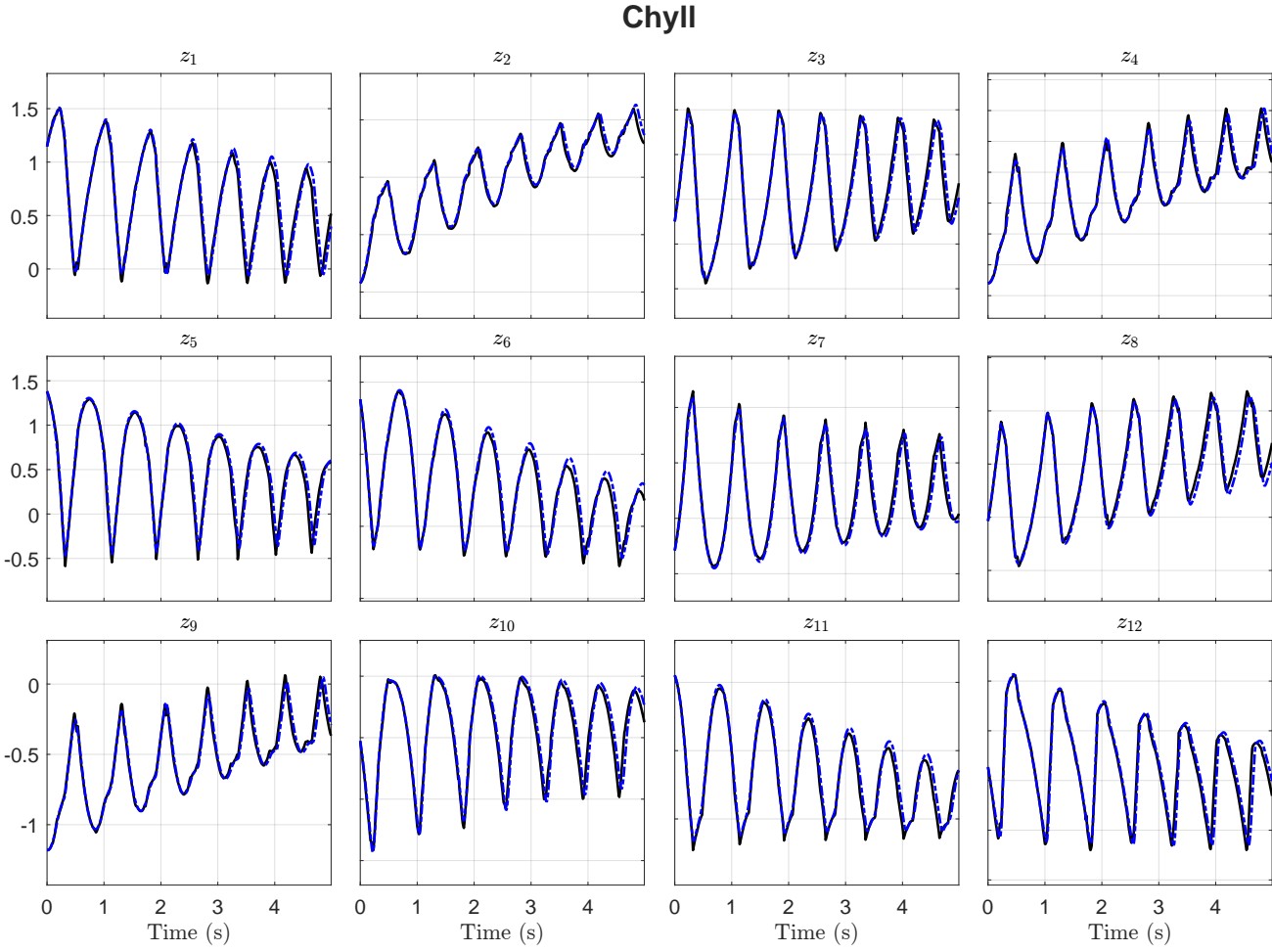

Figure 17: Prediction of latent trajectories of 3D Bouncing Ball for (Teng et al., 2025).

## D. Ablation Studies

We consider the identical number of layers and width in Section C.1 for all our ablation studies.

We consider $w_x = w_z = 10$ if the corresponding loss is activated. We consider $w_c = 1000$ and $\Lambda = 0.09$ if activated.

We consider $w_g = 1$ and $w_v = 0.1$ if activated. For 3D Bouncing Ball case, $w_g = 1000$ if activated.

For the ablation about the activation functions, we consider the last as $\sin$ or ReLU.

