# OpenReview forum: "Embedding Hybrid Systems into Continuous Latent Vector Fields"
_ICML.cc/2026/Conference — ICML 2026 regular_

### Official Review · Reviewer_U7Zu · 2026-03-07

**Soundness:** 2
**Presentation:** 3
**Significance:** 3
**Originality:** 3
**Overall Recommendation:** 4
**Confidence:** 3

**Summary:**

This paper theoretically proves that an n-dimensional hybrid system can be continuously embedded into an m-dimensional space (m>2n), making it suitable for gradient-based learning. It proposes a corresponding latent Neural ODE framework that uses consistency losses in both latent and state spaces to learn from time-series data. Experiments show the method outperforms existing approaches in learning hybrid dynamics.

**Compliance With Llm Reviewing Policy:**

Affirmed.

**Final Justification:**

This paper provides a rigorous theoretical proof for the continuous embedding of hybrid systems and supports it with a clear and effective algorithmic framework. The authors' rebuttal addressed my doubts. It would be even better if this could be validated in a real-world system in the future.

**Key Questions For Authors:**

1.“However, a randomly initialized neural event function is generally ill-conditioned, making the simulation difficult to proceed."
Is there more detailed evidence or theoretical analysis to support this point?
2. Could you provide some more specific examples of hybrid systems, including an explanation of why they are considered hybrid?
3. What is the main difference between your work and (Teng et al., 2025) ?
4. It seems there is a lack of experiments on real-world datasets. Considering sveral examples of hybrid systems you listed (legged locomotion, motion planning, etc.),  what are the main factors that constrain the practical application of your method?

**Limitations:**

yes

**Strengths And Weaknesses:**

This paper presents a rigorous theoretical proof for the continuous embedding of hybrid systems and backing it with a clear, effective algorithmic framework. Its main strength lies in its significant originality, theoretically justifying and empirically demonstrating a superior approach to learning discontinuous dynamics via continuous latent spaces.
A relative weakness is that the practical necessity of the proven velocity continuity condition is not fully established, as the ablation study shows the corresponding loss term provides little benefit. Overall, the work makes a solid and noteworthy advance in making hybrid systems tractable for modern gradient-based learning.

---

> ### Author Rebuttal · Authors · 2026-03-30
>
> We thank the author for the constructive feedback and excellent questions.
>
> **Necessity of the proven velocity continuity condition:**
>
> (Teng et al., 2025) applies the classical Whitney Embedding Theorem to show that the latent space can be continuous. However, there is **no argument on the condition** when the latent vector field is continuous. We believe the continuity of the vector field is important to further improve the numerical well-posedness for learned latent dynamics, as an MLP can not uniformly approximate discontinuous functions without discontinuous components.
>
> To fill this gap, we explored the **dimension bound**. Our ablation still shows that the dimension bound is necessary, while the **specific loss $\mathcal{L}_v$** does not provide much improvement. This can be justified by the fact that the latent vector field NN is by-construction Lipschitz continuous.
>
> **Q1-Ill conditions of event function:**
>
> The following hybrid system theory paper lists a few sufficient conditions to avoid the ill conditions.
>
> *Simic, S.N., Johansson, K.H., Lygeros, J. and Sastry, S., 2005. Towards a geometric theory of hybrid systems. Dynamics of Continuous, Discrete and Impulsive Systems Series B: Applications and Algorithms, 12(5-6), pp.649-687.*
>
> One example of the violation is that the flow at the post-reset states is pointing outward of the guard surface, which will make the state enter another domain immediately when the state enters the domain, which will result in undesired chattering around the guard surface. This effect will make the event-based simulation hard to proceed and is not aligned with many physical systems.
>
> For a neural event function, the randomly initialized weights have no guarantee that these sufficient conditions hold. Empirically, in our simulations, we frequently see the simulation stuck due to infinitely many state changes at the same time.
>
> Additionally, the Zeno behaviour can happen in a neural event function that also makes the simulation hard to continue.
>
> **Q2-Example of Hybrid Systems:**
>
> One example is the humanoid robot. When the robot steps on the ground, the generalized velocity of each joint will change abruptly. The abrupt velocity change can not be modeled by a continuous vector field, while a reset function can well describe it.
>
> Another example is the robot Task and Motion Planning (TAMP). In TAMP, each long-time task is decomposed into multiple modes. The transition between modes are discrete state reset, while inside each mode, the robot is governed by continuous-time dynamics.
>
> In general, a broad class of systems with a mixture of continuous-time and discrete-time dynamics is suitable to be modeled by hybrid systems.
>
> **Q3-Comparison with (Teng et al. 2025):**
>
> The Chyll framework in the (Teng et al. 2025) paper is decoder-free when training the dynamics/encoder networks. Like any decoder-free framework, Chyll needs to implicitly encourage the network to be non-degenerate by some regularizers, but this is not always guaranteed.
>
> The proposed work trains the decoder/encoder/dynamics at the same time, which has a better reconstruction and is not degenerate in the 3D bouncing ball cases.
>
> **Q4-Real World Experiment:**
>
> The main theory is applicable to **deterministic** hybrid systems with finite dimensions. For realworld dataset, the main difficulty is the **partial observability**, **stochasticity**, and **nonlinearity**.
>
> This work solves the non-differentiability of the hybrid event, while future work will tackle the mentioned three issues.
>
> For real-world autonomous driving, the vehicle is more suitable to be modeled as a stochastic hybrid automaton instead of a deterministic one. In the stochastic case, the underlying topology does not hold as the state reset is a probability distribution instead of a deterministic one-to-one mapping. Thus, we need a different framework other than the Whiteney type argument.
>
> An extension considering the following theoretical foundation will be helpful to solve this problem:
>
> *Hu, J., Lygeros, J. and Sastry, S., 2000, March. Towards a theory of stochastic hybrid systems. In International Workshop on Hybrid Systems: Computation and Control (pp. 160-173). Berlin, Heidelberg: Springer Berlin Heidelberg.*
>
> For partial observable systems, the lack of observation will introduce observability problems. The fundamental problem is to identify $x$ from $y=h(x)$ where **dim y < dim x**. The information carried by $y$ can be insufficient to determine the dynamics.
>
> For high-dimensional hybrid systems, the proposed method needs more inductive bias to improve the reconstruction accuracy. For a high-dimensional contact-rich robotics system, such as a humanoid, encoding the integrator structure of joint position and joint velocity to the learning pipeline will be useful.
>
> We consider the three directions as possible future directions to make the proposed method work on real-world datasets.

---

> > ### Author Rebuttal · Reviewer_U7Zu · 2026-04-02
> >
> > Thank you for your detailed and thoughtful response to my previous comments. I appreciate the effort you have put into addressing the concerns raised. Having carefully reviewed your rebuttal, I am inclined to maintain my original score. That said, I believe the work would be further strengthened if you could provide additional validation on more complex, real-world problems beyond the current benchmarks. Such evidence would help better demonstrate the practical impact and robustness of your method.

---

> > > ### Author Response · Authors · 2026-04-06
> > >
> > > We thank the reviewer again for the time and constructive feedback!
> > >
> > > We will address the points mentioned here in our future work and clarify them in the accepted version!

---

### Official Review · Reviewer_kXW7 · 2026-03-08

**Soundness:** 2
**Presentation:** 2
**Significance:** 3
**Originality:** 2
**Overall Recommendation:** 4
**Confidence:** 2

**Summary:**

This paper proves that hybrid dynamical systems, despite their discontinuities, can be represented as continuous dynamics in a higher-dimensional latent space. Based on this result, it introduces a Neural ODE framework that learns this latent embedding and achieves strong performance on hybrid systems with impacts and contact dynamics.

**Compliance With Llm Reviewing Policy:**

Affirmed.

**Final Justification:**

The rebuttal has addressed my main cncerns and changed my evaluation to 4.

**Key Questions For Authors:**

1. The ablation study shows that the velocity compatibility loss L_v has a negligible impact on performance. The paper argues this is because the Lipschitz continuity of the network already enforces the condition. Does this imply that the velocity compatibility condition  in Theorem 6, while theoretically important, is practically redundant when using neural network function approximators?

2. Regarding the discontinuous decoder: If the goal is to perform control or planning in the continuous latent space, this discontinuity seems like a major hurdle for mapping plans back to the physical world. Have you explored any concrete solutions, such as using a decoder that outputs a distribution over possible physical states?

**Limitations:**

Yes.

**Strengths And Weaknesses:**

Strength: The main theorem is a non-trivial result that provides a new perspective on learning hybrid systems. By proving that a continuous representation is generically possible, the paper shifts the problem from ad-hoc architectural design to a well-posed geometric learning task. The proposed learning algorithm flows from the theory. The loss function, with its terms for state consistency (L_x), latent consistency (L_z), gluing (L_g), and velocity compatibility (L_v), is a direct translation of the theoretical requirements into a practical optimization objective.

Weakness:
1. While the benchmarks are well-chosen, they are all low-dimensional ODE systems. The paper does not test the approach on any high-dimensional, spatio-temporal systems (PDEs) that exhibit hybrid dynamics. Please comment.

2. As the authors identify in Section 8, if the encoder maps two distinct physical states to the same latent point to ensure continuity, the decoder must necessarily be discontinuous. The paper acknowledges this but does not offer a solution, suggesting atlas learning or step functions as future work. Please comment on this.

---

> ### Author Rebuttal · Authors · 2026-03-30
>
> We thank the author for the constructive feedback and excellent questions.
>
> **High-dim / PDE systems:**
>
> We agree with the reviewer that the current experiments are limited to low-dimensional ODEs. This is an important limitation of the current empirical section.
>
> Our initial goal in this paper is to propose Theorem 6 and validate it. Low-dimensional ODE systems provide a controlled testbed for this purpose.
>
> Theorem 6 can apply to any finite-dimensional ODE. However, learning higher-dimensional systems is challenging due to the stiffness of the systems and data coverage, which can not be solved properly without more system inductive bias. We will mention these limitations and explore if more inductive bias can fix this issue in future work.
>
> Given proper discretization, PDEs can be transformed to ODEs where Theorem 6 can apply. The main difference lies in the boundary conditions. We will clarify in the paper that PDE-scale validation is beyond the scope of the current submission and is an important direction for future work.
>
> **Discontinuity:**
>
> We agree this is not a trivial problem.
>
> In our current implementation, we are using a continuous neural network to approximate the discontinuous decoders. Thus, the approximation is only good in terms of L-1 or L-2 norm in function space.
>
> However, the proposed method, such as Atlas learning or MLP with step function, is **fundamentally capable** of expressing the discontinuous function. The key is to introduce discontinuity and not sacrifice the universal approximation property. If the Atlas learning has a hard chart switch and the step function is applied in a neural network, there is no theoretical obstacle to learning the discontinuous decoder.
>
> **Q1-Redundancy of Theorem 6:**
>
> Theorem 6 is an existence result suggesting that the induced vector field $Df(x)V(x)$ by the encoder $f$ can be continuous **as long as $m > 2n$**. This is not an off-the-shelf theorem like the Whitney Embedding Theorem, so we investigate this argument and make this proof. The dimension bound $m>2n$ is the core of Theorem 6.
>
> If we are not aware of the dimension bound $m > 2n$, we may choose a mapping $f: M\rightarrow \mathbb{R}^m$ with $m < 2n$ to enforce $Df(x^+)V(x^+) = Df(x^-)V(x^-)$, which can fail due to the topological obstructions. Thus, Theorem 6 is not redundant, but very necessary to show that such a continuous structure exists.
>
> Another implication of Theorem 6 is that in a higher-dimensional space, the vector field is not only piecewise continuous but also continuous. This will guide the world model design and learning for dynamics to show how the latent is **well-posed by construction**.
>
> What our ablation suggested is that **this specific loss** $\mathcal{L}_v$ is not necessary. On the one hand, the neural network is **by-default** continuous. On the other hand, the velocity approximated by finite difference may not be accurate enough, while it is a well-adopted choice. Though this result is not positive, we believe it is worth discussing, as many structure-inspired works tend to add structures, while this is not our case.
>
> **Q2-Latent space planning:**
>
> This is an excellent question!
>
> If the encoder is an embedding, such that $f(x) = f(y) \Rightarrow x=y$, any goal-reaching task can be defined in the latent space $\mathrm{Img} f(\cdot)$, as the latent state **uniquely** specifies the state in the original state space. In other words, the planning can be decoder-free. In our hybrid system case, the latent dimension is higher, and by Theorem 6, it is generically an embedding. Additionally, we do not need the decoder if the reward is also well-defined using the latent state alone.
>
> Examples include:
>
> *Sobal, V., Zhang, W., Cho, K., Balestriero, R., Rudner, T.G. and LeCun, Y., Learning from Reward-Free Offline Data: A Case for Planning with Latent Dynamics Models. In The Thirty-ninth Annual Conference on Neural Information Processing Systems.*
>
> *Hansen, N., Su, H., and Wang, X., TD-MPC2: Scalable, Robust World Models for Continuous Control. In The Twelfth International Conference on Learning Representations.*
>
> If recovering the state in the state space is required, we need to calibrate the distribution of the decoder output, which can have large errors around the switching surface. In this case, a generative decoder can be a better choice. One possible solution is to parameterize the decoder as a diffusion model, which will be able to generate the decoded state given the uncertainty / multi-modality of the decoder around the guard surface.
>
> In such a stochastic case, the following stochastic hybrid system framework is also a good choice, such as the following work:
>
> *Hu, J., Lygeros, J. and Sastry, S., 2000, March. Towards a theory of stochastic hybrid systems. In International Workshop on Hybrid Systems: Computation and Control (pp. 160-173). Berlin, Heidelberg: Springer Berlin Heidelberg.*

---

> > ### Author Rebuttal · Reviewer_kXW7 · 2026-04-04
> >
> > I thank author's feedback, I would like to adjust my score accordingly.

---

> > > ### Author Response · Authors · 2026-04-06
> > >
> > > We thank the reviewer again for the time and constructive feedback!

---

### Official Review · Reviewer_eoUB · 2026-03-08

**Soundness:** 4
**Presentation:** 4
**Significance:** 4
**Originality:** 4
**Overall Recommendation:** 5
**Confidence:** 3

**Summary:**

The paper shows that a so-called \emph{hybrid system} of dimension $n$ can be embedded into a euclidean space of dimension $m>2n$. This establishes an appropriate Whitney-style embedding theorem for hybrid systems, but in a way where the embedding additionally respects the dynamics of the hybrid system.

**Compliance With Llm Reviewing Policy:**

Affirmed.

**Final Justification:**

I appreciate the response in the rebuttals, which have reaffirmed the score of accept. I have raised my confidence score to 3.

**Key Questions For Authors:**

My main question for the authors does not relate to the current paper \emph{per se}, but an extension of their main theorem and I am curious what their thoughts on it are.

The construction in the proof of Theorem 6 is inherently \emph{non-constructive} as it relies on Baire-category arguments. Is there a constructive version? The specific question I have in mind for this is the following:

1. Suppose $\mathcal{R}=(\mathbb{R};<,+,\cdot,\ldots)$ is an o-minimal expansion of the real field, and $\mathcal{H}=(M,S,V,r)$ is a hybrid system such that $M\subseteq\mathbb{R}^q$ is a definable embedded smooth submanifold, and the data $S,V,r$ is likewise also definable in $\mathcal{R}$. Then can the $f$ in Theorem 6 be chosen so that it is also definable in $\mathcal{R}$?

Hybrid systems were first considered in the o-minimal context in \emph{O-minimal hybrid systems} by Lafferrier, Pappas, Sastry (2000).

Another question along these lines is the following:

2. Is there a good example of a hybrid system which is not definable in an o-minimal structure?

**Limitations:**

yes

**Strengths And Weaknesses:**

The main strength of the paper is that it establishes a foundational embedding result for hybrid systems analogous to the Whitney embedding theorem for manifolds. Such a result has a place in the "first chapter" of the theory of hybrid systems, and I am surprised it was apparently not previously known.

There are no major weaknesses from what I can tell, although some definitions are at times unclear (for a non-expert like myself). For example:
* In 3.1 and 3.2, is a \emph{smooth manifold} always a \emph{smooth manifold with boundary}? The way Definition 3 is written suggests the answer is "yes", whereas the fact that Theorem 1 explicitly mentions "with boundary" suggests the answer is "no".
* Remark 1 is more of a definition for the paper than a remark: we say a property of points in a topological space is \emph{generic} if it is large in the sense of Baire-category, i.e., if it holds for all points on a \emph{residual} (aka. \emph{comeager}) set. Recall that a subset of a topological space is \emph{comeager} if its complement is a countable union of nowhere dense sets. [I think the footnote should say "unions" instead of "intersections".]

---

> ### Author Rebuttal · Authors · 2026-03-30
>
> I would like to thank the author for the appreciation of the technical contribution of this work.
>
> **Clarity of definition 3:** By definition, $S$ is the guard surface that $S\in \partial M$. Thus, in Def. 3, $M$ is a manifold with boundary. We will make this clear in the accepted version by explicitly mentioning that $M$ is a smooth manifold with boundary to avoid possible ambiguities.
>
> **Clarity of Remark 1:** We will consider renaming this remark as a definition or making it more remark-like in the accepted version. We will correct the *intersections* to *unions* in the footnotes, and we thank the author for noticing this issue.
>
> **Q1-Constructive version:**
>
> Thank you for the excellent questions! Our current work mainly considers the geometric structure of the hybrid systems, while the algebraic structure provides more insights into computation.
>
> Yes, the construction in Theorem 6 is generic and not a refinable construction.
>
> In Sec. 7.1, we provided a constructive example with an analytical solution, which is a one-dimensional hybrid system lifted to a two-dimensional space.
>
> For the general case, we may need to consider the proof of Theorem 6 using the Parametric Transversality Theorem (PTT) with additional o-minimal structure. The current generic structure can not say if the encoder is also definable. Possible reference showing the counterpart of PTT with the **definable topology**:
>
> *Nguyen, N. and Trivedi, S., 2020, May. Transversality of smooth definable maps in O-minimal structures. In Mathematical Proceedings of the Cambridge Philosophical Society (Vol. 168, No. 3, pp. 519-533). Cambridge University Press.*
>
> Whether $f$ is definable can be investigated by considering such a topology.
>
> We believe an analytical solution to a definable $f$ will be hard. Though constraining the boundary condition is not too complicated, the construction of the global injectivity/immersion will be hard, as our construction can only be done in finite dimensional space. The sum-of-squares optimization can be a good tool to obtain such a construction.
>
> **Q2-Nondefinable Hybrid Systems:**
>
> One example we can consider is a 1-D system with guard $\frac{1}{N}, N=... -3, -2, -1, 1, 2,...$. In this case, the open subset with $0$ in its closure will have infinitely many elements, which is not a finite set.
>
> **Confidence:**
>
> Your assessment of our use of Baire-category arguments is accurate. The main body of the proof contains two constructions. In B.1, we constructed the tangential and normal components of the vector field on the embedded images of the guard. And then we apply the Parametric Transversality Theorem (PTT) to show the dimension bound. In B.3, we constructed the embedded images in the entire domain, and then followed the PTT. To accurately describe the geometry of hybrid systems, B.1 constructs everything in the Collar embedding, which involves more differential topology.
>
> We hope this explanation can increase the clarity of this work, and we will include more explanations of the main proof in the accepted version.
>
> Based on your insightful comments, **we believe your expertise in hybrid system theory/topology is excellent**. While we appreciate your humility, we believe your confidence level can be higher, and we wonder if you could consider **adjusting your confidence to a higher score**, such as **4** or **5**.

---

> > ### Author Rebuttal · Reviewer_eoUB · 2026-04-02
> >
> > The rebuttal has addressed my concerns. I thank the authors for pointing to the Nguyen and Trivedi reference, and considering my question. I will raise my confidence score modestly to 3.

---

> > > ### Author Response · Authors · 2026-04-06
> > >
> > > We thank the reviewer again for the time and constructive feedback!
> > >
> > > We appreciate the reviewer for the in-depth questions and **raising the confidence score**.

---

### Official Review · Reviewer_WYL2 · 2026-03-13

**Soundness:** 3
**Presentation:** 3
**Significance:** 3
**Originality:** 3
**Overall Recommendation:** 5
**Confidence:** 1

**Summary:**

This paper converts a discontinuous hybrid dynamical system into a continuous latent dynamical system and learns that latent dynamics with a Neural ODE rather than explicitly modeling discrete modes or event functions. It treats reset-linked states as equivalent points on a quotient manifold called the hybrifold $M_H = M/\sim$ and proves that for a hybrid system of dimension $n$, there generically exists an encoder $f: M \rightarrow \mathbb{R}^m$ with latent dimension $m > 2n$ that maps reset-paired states to the same latent point while aligning their pushed-forward velocities, making the latent trajectory continuous despite jumps in the original state. Based on this result, the model consists of an encoder $f_\theta$, a latent vector field $V_\theta$, and a decoder $f_\theta^{-1}$, which encodes an initial state, evolves it with a Neural ODE in latent space, and decodes predictions back to the original space, respectively. In the training, the model uses 5 losses: reconstruction loss $\mathcal{L}_x$, latent consistency loss $\mathcal{L}_z$, gluing loss $\mathcal{L}_g$ for pre/post-reset pairs, optional velocity-matching loss $\mathcal{L}_v$, and covariance loss $\mathcal{L}_c$ to prevent latent collapse. During execution, the model employs a rollout curriculum that gradually increases prediction horizon. The authors perform experiments on different systems and show improved performance over baselines. Additionally, they perform ablation studies on the losses and show that latent consistency $\mathcal{L}_z$ is most critical, $\mathcal{L}_g$ and $\mathcal{L}_c$ provide modest gains, and explicit velocity matching $\mathcal{L}_v$ often offers little benefit.

**Compliance With Llm Reviewing Policy:**

Affirmed.

**Final Justification:**

I will keep my high score.

**Key Questions For Authors:**

Approximately how much computation is required for the proposed method?

**Limitations:**

The authors did not put the required “Impact Statement” section in the paper.

**Strengths And Weaknesses:**

This paper generally looks well written, with technically sound theoretical results, good comparison with appropriate baseline methods, and thorough ablation studies. Simulation on more complicated systems will be beneficial.

---

> ### Author Rebuttal · Authors · 2026-03-30
>
> I would like to thank the author for the appreciation of the technical contribution of this work.
>
> **Computation:**
>
> The computation is not high for our cases. The model parameters are mentioned in the Appendix. C.
>
> Our method considers the MLP for the 3D bouncing ball example, which has the largest number of parameters, which is
>
> Vector field: [256]x2 (Proposed)
>
> Encoder:[256]x2 (Proposed)
>
> Decoder: [256]x3 (Proposed)
>
> The other cases consider even smaller networks
>
> **Impact Statement:**
>
> Sorry for this problem. We will add the following content in the accepted version:
>
> *This paper presents work whose goal is to advance the field of machine learning. There are many potential societal consequences of our work, none of which we feel must be specifically highlighted here.*

---

> > ### Author Rebuttal · Reviewer_WYL2 · 2026-04-03
> >
> > The authors provided a response to my question.

---

> > > ### Author Response · Authors · 2026-04-06
> > >
> > > We thank the reviewer again for the time and constructive feedback!

---

### Decision · Program_Chairs · 2026-04-30

**Decision:**

Accept (regular)

**Comment:**

This paper makes a solid contribution to the community by establishing a generic continuous embedding result for hybrid systems and by translating that insight into a latent Neural ODE framework with strong performance on benchmark problems. The reviewers were broadly positive about the novelty and technical significance of the main theorem, though enthusiasm about the empirical scope was more moderate. In the author–reviewer discussion, the authors satisfactorily clarified the geometric assumptions, the role of the velocity continuity condition, the distinction from prior work, and practical issues such as event-function ill-conditioning and decoder discontinuity. Several reviewers explicitly stated that their concerns were resolved, and some increased their confidence or overall assessment after the rebuttal. The main remaining limitation is that the experiments are still confined to relatively low-dimensional synthetic systems, so validation on more complex or real-world hybrid systems remains future work. Overall, the theoretical contribution is nontrivial, the method is well aligned with the theory, and the rebuttal strengthened confidence in the paper. I recommend acceptance.